# Rectifying disorder of extracellular matrix to suppress urethral stricture by protein nanofilm-controlled drug delivery from urinary catheter

Juanhua Tian[1], Delai Fu[1], Yongchun Liu[2], Yibing Guan[3], Shuting Miao[2], Yuquan Xue[1], Ke Chen[4], Shanlong Huang[1], Yanfeng Zhang[5], Li Xue[1], Tie Chong [1] ✉ & Peng Yang [2,6,7] ✉

Urethral stricture secondary to urethral injury, afflicting both patients and urologists, is initiated by excessive deposition of extracellular matrix in the submucosal and periurethral tissues. Although various anti-fibrotic drugs have been applied to urethral stricture by irrigation or submucosal injection, their clinical feasibility and effectiveness are limited. Here, to target the pathological state of the extracellular matrix, we design a protein-based nanofilm-controlled drug delivery system and assemble it on the catheter. This approach, which integrates excellent anti-biofilm properties with stable and controlled drug delivery for tens of days in one step, ensures optimal efficacy and negligible side effects while preventing biofilm-related infections. In a rabbit model of urethral injury, the anti-fibrotic catheter maintains extracellular matrix homeostasis by reducing fibroblast-derived collagen production and enhancing metalloproteinase 1-induced collagen degradation, resulting in a greater improvement in lumen stenosis than other topical therapies for urethral stricture prevention. Such facilely fabricated biocompatible coating with antibacterial contamination and sustained-drug-release functionality could not only benefit populations at high risk of urethral stricture but also serve as an advanced paradigm for a range of biomedical applications.

Urethral stricture is a common disease (200–1200 cases per 100,000 individuals[1]) following urethral injury. The pathological state of extracellular matrix (ECM) metabolic disorder caused by injury (including surgery, pelvic fracture, inflammatory injury, and traumatic catheterization) results in the replacement of normal urethral tissue by dense fibers interspersed with fibroblasts[1]. This aggravated fibrosis eventually leads to progressive urethral lumen reduction, consequent symptomatic dysuria, and even renal impairment. With the growing

[1]Department of Urology, The Second Affiliated Hospital of Xi'an Jiaotong University, West Five Road, No. 157, 710004 Xi'an, China. [2]Key Laboratory of Applied Surface and Colloid Chemistry, Ministry of Education, School of Chemistry and Chemical Engineering, Shaanxi Normal University, 710119 Xi'an, China. [3]Department of Urological Surgery, The First Affiliated Hospital of Zhengzhou University, 450052 Zhengzhou, Henan Province, China. [4]Key Laboratory of Bio-Inspired Smart Interfacial Science and Technology of Ministry of Education, School of Chemistry, Beihang University (BUAA), 100191 Beijing, China. [5]School of Chemistry, Xi'an Jiaotong University, 710049 Xi'an, China. [6]International Joint Research Center on Functional Fiber and Soft Smart Textile, School of Chemistry and Chemical Engineering, Shaanxi Normal University, 710119 Xi'an, China. [7]Xi'an Key Laboratory of Polymeric Soft Matter, School of Chemistry and Chemical Engineering, Shaanxi Normal University, 710119 Xi'an, China. ✉e-mail: chongtie@126.com; yangpeng@snnu.edu.cn

demand for healthcare brought about by the aging population and advances in medical technology, the increasing incidence of mucosal injury and secondary urethral strictures caused by various transurethral procedures has attracted great attention[2,3]. Especially in patients with spinal cord injury or in intensive care, urethral stricture or erosion due to long-term catheterization has been reported as high as 8.7%[4]. Once urethral injury progresses to urethral stricture, subsequent treatment is extremely troublesome, which will pose challenges to both patients and urologists. At present, the long-term success rate of the most commonly used endoscopic treatment is only 20–30%[1], and although urethroplasty has developed rapidly in recent years, fibrous scarring may still develop in the urethral submucosa after substitute surgery, and this open surgery is less suitable for the elderly and frail patients. However, the pathological state of ECM metabolic disturbance and subsequent scar repair is not be rectified regardless of the treatment[5]. Therefore, there is an urgent need to develop alternative strategies other than surgery and attempt to focus on regulating healing by inhibiting the progression of fibrosis following urethral injury to prevent urethral strictures.

Although systemically administered anti-fibrotic agents have shown efficacy in attenuating tissue/organ fibrosis in animal models[6,7], off-target side effects have greatly limited their success in clinical trials[8]. By contrast, the successful attempt of a paclitaxel-coated balloon combining mechanical dilation and local drug delivery in the treatment of recurrent urethral strictures suggests the feasibility and promise of local therapy to interfere with urethral strictures[9]. Currently, in order to inhibit secondary strictures after urethral injury, various anti-fibrotic drugs have been administered locally by hydrostatic pressure, submucosal injection, urethral irrigation, drug-eluting stents, and catheters[10-14]. However, most of these strategies show limited efficacy and practicality in clinical due to poor local drug retention, possible intractable complications, and high demands on patient compliance[15,16]. Furthermore, there are substantial scientific challenges for the construction and further application of anti-fibrotic implants (e.g., catheters and stents), mainly including the difficult

integration of a robust delivery platform with a catheter[17], the limited drug loading capacity[18], mechanical instability of the delivery system and suboptimal release behavior[11,19]. As a result, the initial burst drug release may expose the body to excessive drug doses and possible related side effects, while premature depletion of drug reserves makes it difficult to guarantee an effective dose in the later period. More importantly, refractory urinary tract infections caused by bacterial colonization and biofilm on the catheter surface can also exacerbate urethral fibrosis and lead to surgical failure by triggering an inflammatory response[20]. Taken together, despite the urgent clinical need, it remains a formidable challenge to develop anti-fibrotic implants with several superior features to overcome the above clinical drawbacks. To meet these challenges, using amyloid-like protein aggregation that can be functionalized virtually arbitrary material surface to combine surface modification with drug delivery might be a good strategy to overcome the limitations associated with the intraurethral administration route.

In this study, we propose a strategy to impart both anti-fibrotic and anti-biofilm activity to the catheter (in principle, being extendable to any other implants) by forming protein nanofilm coating on implant surfaces to noticeably control anti-fibrotic drug release, regulate tissue repair and finally prevent urethral stricture. We utilize unfolded bovine serum albumin (BSA) triggered by tris(2-carboxyethyl)phosphine (TCEP) to assemble a biocompatible protein nanofilm on the catheter. With effective doping sodium alginate (SA) and introducing a sandwich-like structure design, the resultant hybrid PTB@SA nanofilm with reliable toughness could serve as both a modified coating against bacterial biofilms and a reservoir system with a constant activity source to delivery drugs in a controlled manner for tens of days. The rabbit urethral injury model determined that local delivery of an anti-fibrotic drug (rapamycin) from the catheter decreased the lumen reduction from 51.8 to 10.8% by balancing the deposition and degradation of ECM during injury healing (Fig. 1). This anti-fibrotic catheter with a high effect/dose ratio requires only 10.7 and 2.2% of the amount of rapamycin used for urethral irrigation and systemic administration,

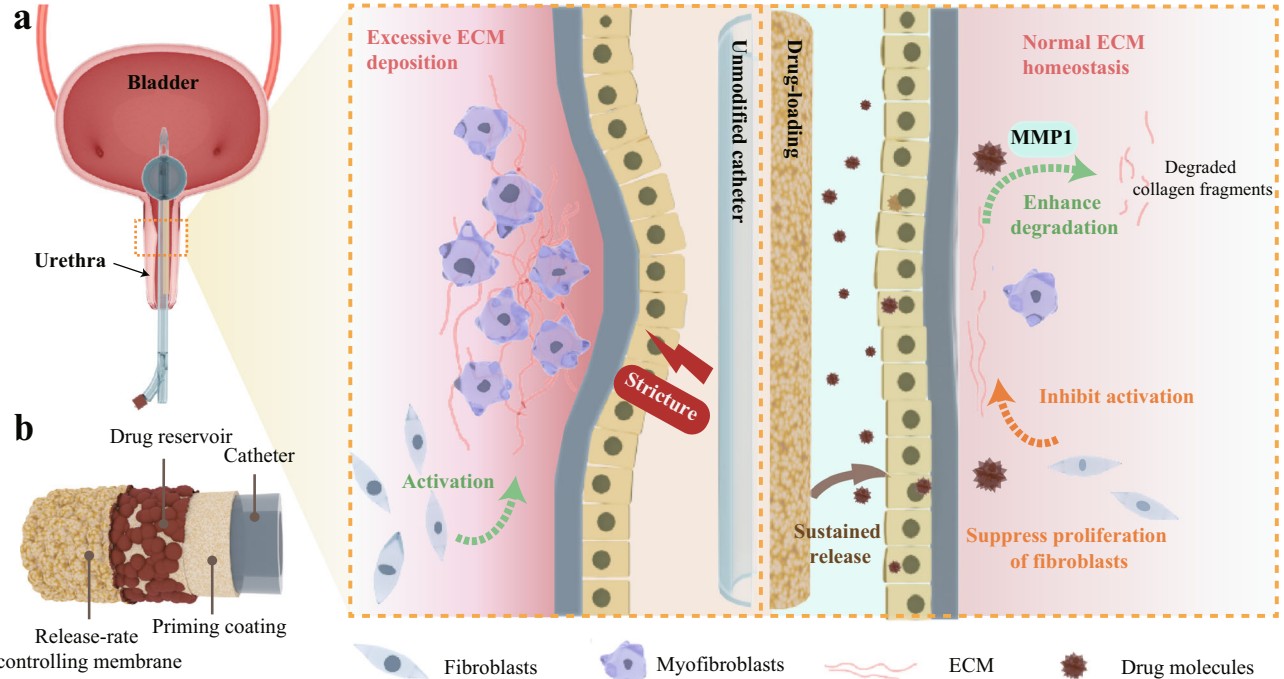

**Fig. 1 | Schematic illustration of the sandwich-like drug delivery platform mounted on the urinary catheter for efficient prevention of urethral stricture.** **a** Sustained release of the specific anti-fibrotic agent from the drug delivery system maintains normal extracellular matrix (ECM) homeostasis by modulating ECM metabolism, thereby inhibiting post-injury urethral stricture. **b** The schematic diagram shows the sandwich-like structure of the drug-loaded catheter.

and further achieves a percent improvement in lumen reduction of 79.2%, which is also the best efficiency reported so far in preclinical studies of topical therapy to prevent urethral stricture[11–14]. Furthermore, the off-target effects related to systemic administration, additional clinical procedures, and the associated complications are circumvented tactfully. Our study not only provides a beneficial prophylactic option for a high-risk group of urethral strictures who have undergone traumatic catheterization or transurethral procedures but also can be further extended to the management of other body cavity strictures characterized by ECM metabolic disturbances and fibrosis. In addition, this work also provides a proof-of-concept for an extensible drug delivery system based on a simple protein nanofilm for implant surface functionalization.

## Results and discussion
### The one-pot synthesis of the hybrid PTB@SA nanofilm
Here, we implemented surface modification by simply immersing any shape of the material in a phase transition solution[21], which was obtained by premixing BSA and SA, followed by the addition of TCEP aqueous solution (pH = 4.5) (Fig. 2a). With introducing SA into such phase transition system, the aggregation of BSA triggered by the reduction of the intramolecular disulfide bond of BSA via TCEP proceeded in a similar manner to that without SA[22]. Driven by the hydrophobic interaction, the resultant β-sheet-stacking protein particles with diameters of 12–16 nm self-assembled at the solid–liquid interface to form a nanofilm (Fig. 2b, Supplementary Fig. 1). The geometrically calculated size of the pore surrounded by three nanoparticles inside the hybrid nanofilm is 1.86–2.48 nm[23], which is consistent with the measurement of approximately 2.16 nm as reflected by Brunauer–Emmett–Teller (BET) test (Fig. 2c). These voids among the protein nanoparticles provide intrinsic nanochannels for molecular diffusion[23], which thus serve as the physical foundation for drug loading and delivery in the present work. The change in the main types of BSA secondary structure from α-helix to β-sheet during the phase transition is reflected by circular dichroism (CD) and Fourier transform infrared (FTIR) spectra[24,25] (Fig. 2d, Supplementary Fig. 2). Employing the binding of Thioflavin T (ThT) and Congo red to β-sheet[26,27], we further confirmed the accumulation of β-sheet structure during phase transition and the successful modification of the hybrid coating on the silicone rubber (SR) catheters/sheets (Fig. 2a inset, Fig. 2e). The thickness of the nanofilm can be well controlled in the range of 10–200 nm by adjusting the concentration of BSA, SA additive proportion, the reaction temperature and the incubation time (Supplementary Fig. 3). For simplicity, we named the nanofilm with varying degrees of hybridization as follows: PTB@SA(n), where n is the mass ratio of SA to BSA in the phase transition system.

To directly visualize the hybridization and determine the encapsulation density of SA, 5(6)-aminofluorescein-labeled SA was added to the phase transition solution during the fabrication of the nanofilm. The results confirmed that SA was uniformly distributed in the nanofilm (Fig. 2f), and its encapsulation density also increased correspondingly with the addition ratio (Fig. 2g, Supplementary Fig. 4), indicating that SA was successfully and finely incorporated into the nanofilm. X-ray photoelectron spectroscopy (XPS) wide scan for the PTB@SA nanofilm coating on SR then reflected the absence of phosphorus and silicon, indicating the exclusion of TCEP in the nanofilm and a completely covered coating on the substrate (Supplementary Fig. 5). Further, due to the presence of SA as well as hydrophilic and hydrophobic amino acid residues, the nanofilm surface exhibited abundant functional groups, as determined by XPS. These functional groups, mainly including aliphatic carbon (C–C/C–H), amines (C–N), hydroxyl (C–O), amides (O=C–N), thiols (C–S) and carboxyl (O=C–O) groups[28] (Fig. 2h), provided multiplex interfacial bonding sites for the nanofilm with underlying substrates[29]. Therefore, it enabled modification of the surface of virtually arbitrary material (including metal,

organic or inorganic materials) through ligand bonding, electrostatic interaction, hydrophobic interaction, and hydrogen bonding[21]. Quartz crystal microbalance with dissipation (QCM-D) also reflected such strong interfacial adhesion activity on the Au chip, as the adsorption amount of PTB@SA was significantly higher than that of native BSA (Supplementary Fig. 6). Furthermore, the water contact angle (WCA) measurement of the hybrid nanofilm on the surface of polyether ether ketone (PEEK), SR, polyvinyl chloride (PVC), polycarbonate (PC), polyethylene terephthalate (PET), silicon wafer (Si), aluminum oxide (Al₂O₃), titanium (Ti), and stainless steel (SLL) showed a consistent WCA approximately 80° (Fig. 2i, Supplementary Fig. 7), also implying a successful modification on versatile substrates by the hybrid nanofilm. The endurance of the hybrid coating under stringent application conditions was further evaluated by subjecting the coating with an initial thickness of 50 nm to external harsh environments, typically including polar solvents, surfactants, body fluids, mechanical challenge, or high-temperature steam. The unchanged nanofilm thickness around 50 nm after these treatments indicated the excellent robustness of the hybrid nanofilm, which is greatly important for practical production and application (Fig. 2j, Supplementary Fig. 8).

### The supramolecular interaction in the PTB@SA nanofilm and resultant toughness enhancement
One of the challenges in building membrane-controlled drug delivery systems on flexible implants such as SR is the susceptibility to undesired coating cracks during repeatable deformation, resulting in uncontrolled drug release. Thus, an important consideration for introducing SA into the PTB system lies in the judgment that the flexible SA molecules may result in a favorable improvement in the coating toughness by anchoring to the relatively rigid PTB nanoparticles through intermolecular interactions[30,31]. Actually, protein-polysaccharide interactions offer new opportunities for biomedicine and green chemistry by forming complexes with improved functional performances, typically including stability, emulsification, gelation, and mechanical properties[30–34]. We speculate that the association between unfolded protein chains and SA is mainly through the electrostatic interaction and hydrogen bonding (Fig. 3a). In such a way, SA might modulate the aggregation of BSA by shielding the hydrophobic interaction among the phase-transitioned protein colloids. The resultant increased toughness is highly desirable for suppressing the breaking and detachment of the coating applied on the catheter for drug delivery (see below).

A few previous studies have proven that the negatively charged SA could interact electrostatically with the positively charged microdomains of BSA[32,33]. In the present work, real-time zeta potential analysis showed that the net charge of the PTB and PTB@SA(0.10) colloids gradually tended to zero as the reaction proceeded (Supplementary Fig. 9a), reflecting that the aggregation may be facilitated by weakening electrostatic repulsion among nanoscale protein colloids. In particular, due to the interaction between anionic SA and PTB, the zeta potential of PTB@SA colloids decreased significantly as the mass ratio of SA to BSA increased from 0.10 to 0.25. This change could lead to an enhanced electrostatic repulsion among protein colloids so as to impede further aggregation. Consequently, in contrast to the PTB and PTB@SA(0.10) presenting macroscopic phase separation, the PTB@SA(0.25) produced a homogeneous emulsion without sediments (Supplementary Fig. 9b–d). Besides the electrostatic interaction, various intermolecular hydrogen bonding sites formed between unfolded protein chains and SA were also pronounced due to the existence of abundant polar groups (such as amine, amide, ether bonds, and hydroxyl groups) in the hybrid nanofilm[33,35,36]. Notably, as the additive proportion of SA increased from 0.05 to 0.20, the corresponding FTIR spectra reflected that the amide II peak at 1538 cm⁻¹ of the PTB (the bending vibration of -NH-) was gradually shifted to 1549 cm⁻¹, while the absorption bands

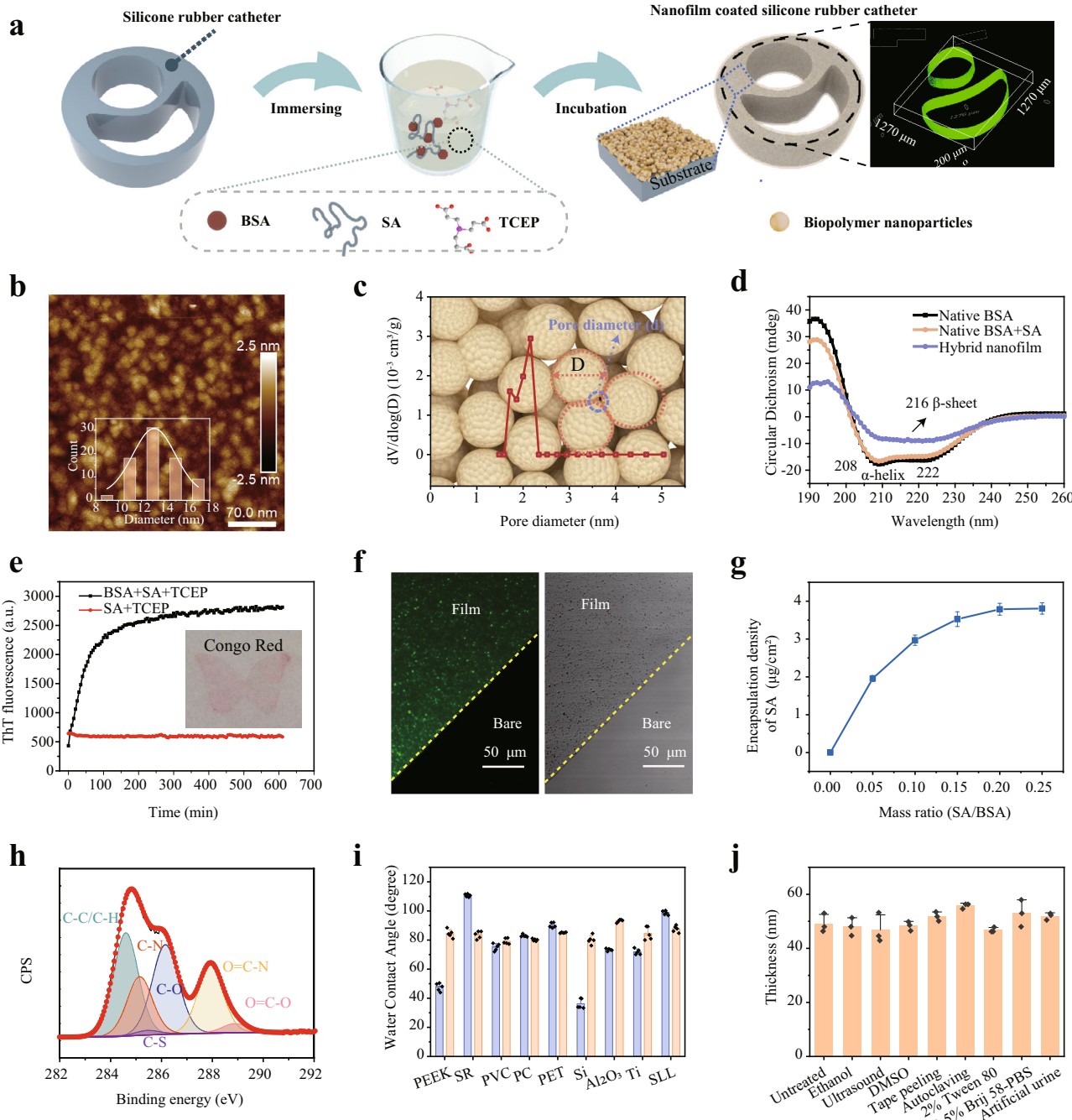

**Fig. 2 | Fabrication and stability of the hybrid coating on general substrates. a** The formation of the hybrid nanofilm on silicone rubber (SR) catheter by the assembly of protein nanoparticles at the solid–liquid interface through dip coating. **b** Atomic force microscopy (AFM) image of the hybrid nanofilm. **c** Pore size analysis of the hybrid nanofilm measured by Brunauer–Emmett–Teller (BET). The background image schematically shows the pores created by the stacking of protein particles from a top view. The pore diameter (d) and particle size (D) have a relationship d = D(2 − √3)/ √3 = 0.1547D. **d** Circular dichroism (CD) spectra of native BSA (single or mixed with SA) and the hybrid nanofilm. **e** Thioflavin T (ThT) fluorescence change as a function of phase transition time, with the insets showing the optical image for Congo-red staining on the nanofilm (the butterfly-shaped

nanofilm prepared by using a corresponding sticker to control the deposition area of the coating). **f** Confocal laser scanning microscope (CLSM) images indicating the even distribution of fluorescently labeled SA in the coating. **g** The encapsulation density of SA across the surface as a function of the amount of SA in the phase transition system. Data are mean ± S.D. $n = 4$ independent samples per group. **h** High-resolution *C1s* deconvolution spectra of the hybrid nanofilm. **i** Water contact angle (WCA) of the nanofilm on various substrates. Data are mean ± S.D. $n = 5$ independent samples per group. **j** Stability evaluation of the nanofilm after different treatments. Data are mean ± S.D. $n = 3$ independent samples per group. Source data are provided as a Source Data file.

associated with C-O-C and -OH/-NH$_2$ stretching vibrations of the hybrid nanofilm red-shifted from 1116 to 1028 cm$^{-1}$, and 3419 to 3280 cm$^{-1}$ respectively, accompanied by an increase in the peak intensity with increasing the ratio of SA (Supplementary Fig. 10). From the deconvolution data in the range of 3000–3700 cm$^{-1}$, it was

further concluded that the proportion of free hydroxyl in the hybrid nanofilm with different SA content was significantly lower than those of pure SA and the mixture of SA and native BSA[32,34,37] (Fig. 3b). The above evidences thus implied that OH···O (ether) and OH···OH were the two main types of hydrogen bonding in the hybrid nanofilm, with

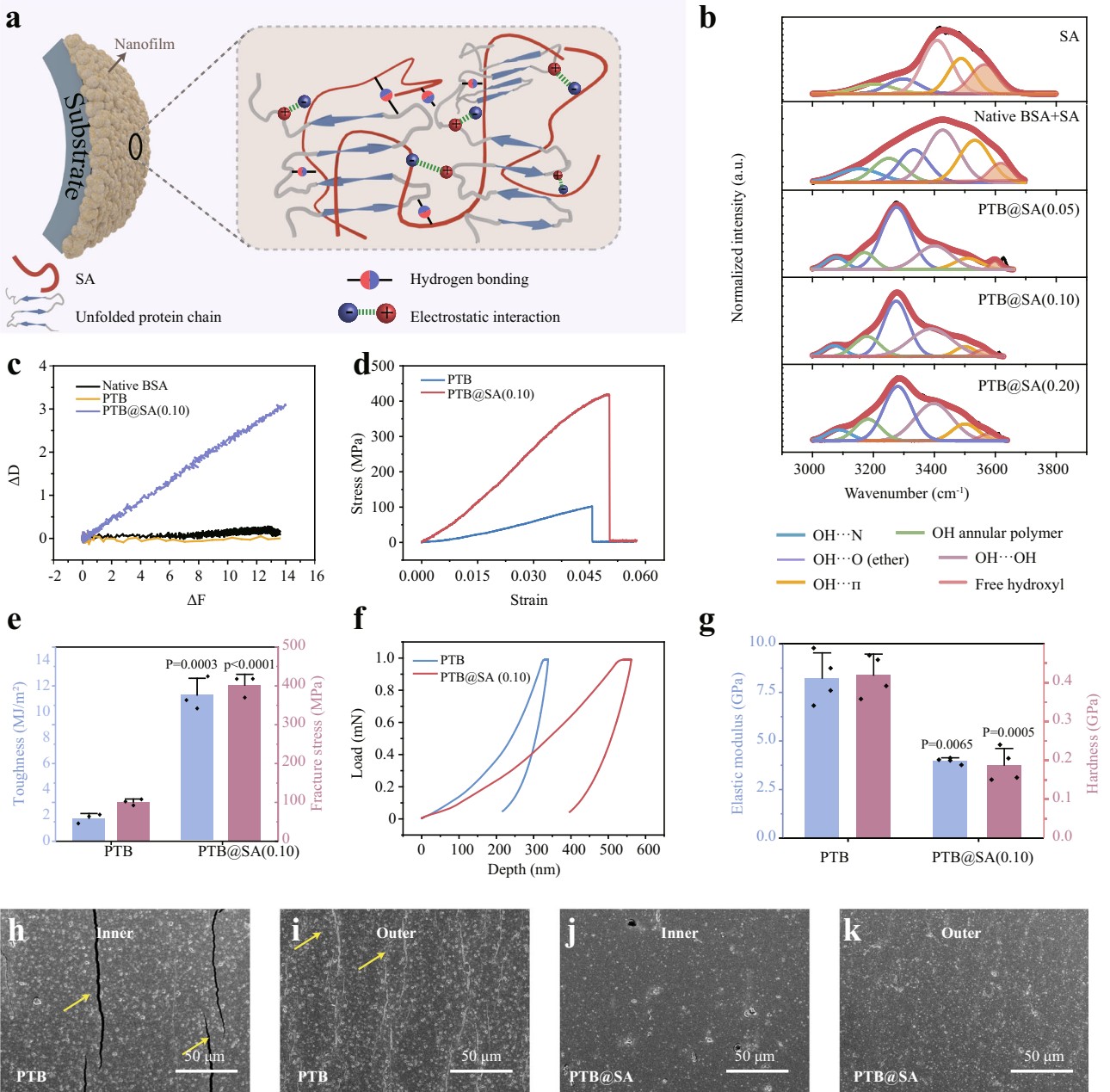

**Fig. 3 | Improved toughness and mechanical stability of the hybrid coating. a** Schematic illustration of supramolecular interactions in the hybrid nanofilm. **b** The deconvoluted IR bands to characterize hydrogen bonding in the hybrid nanofilms. **c** ΔD−ΔF plots from Quartz crystal microbalance with dissipation (QCM-D) indicating the viscoelastic properties of the adsorption layer. **d**, **e** Stress−strain curve (**d**), fracture stress and toughness (**e**) of the PTB and PTB@SA(0.10) nanofilm recorded at 22 °C and 50% relative humidity. Data are mean ± S.D. $n = 3$ independent samples per group. **f**, **g** Loading−unloading curves of PTB and PTB@SA(0.10) nanofilms (**f**) and corresponding elastic modulus and hardness (**g**), as determined by nanoindentation tests. Data are mean ± S.D. $n = 4$ independent samples per group. **h**–**k** Scanning electron microscope (SEM) images of the PTB or PTB@SA(0.10)-modified SR after the fatigue bending test, respectively. Yellow arrows indicate cracks. The experiments in (**h**–**k**) were repeated independently at least three times with similar results. Statistical significance was determined by two-tailed Student's $t$-test (**e** and Hardness in **g**) or two-tailed Welch's $t$-tests (Elastic modulus in **g**). Source data are provided as a Source Data file.

the co-existence of other types of hydrogen bonds such as OH⋯N, OH annular polymer and OH⋯Π (Fig. 3b).

As further demonstrated by QCM-D, the slope of ΔD/ΔF was much steeper for the absorption of PTB@SA(0.10) (0.213) than PTB colloids (0.014) and native BSA (0.008) (Fig. 3c, Supplementary Fig. 11). This result then indicated that through the aforementioned intermolecular interactions, SA made the protein colloids assembled at the interface more viscoelastic and flexible[38]. To further determine the effect of SA in improving the mechanical properties of the coatings, uniaxial tensile and nanoindentation tests were performed. Compared with the PTB

nanofilm, the hybrid nanofilm represented approximately six- and fourfold improvements in toughness and fracture stress, respectively (Fig. 3d, e, Supplementary Fig. 12). Such toughness enhancement was reasonably accompanied by a 50% increase in the indentation depth as well as 52.4% decrease of elastic modulus and 55.8% decrease of hardness (Fig. 3f, g). The reduced elastic modulus and hardness of the hybrid material, as well as its increased fracture stress and toughness, ensured the integrity and stability under bending stress, which is an important evaluation index for the surface coating on flexible silicone substrates. As shown in Fig. 3h–k, the PTB@SA(0.10) coating on the SR

remained intact after 300 times of internal or external bending with a bending radius of 2 cm (Supplementary Fig. 13, 14), while deep and wide cracks appeared on the pure PTB coating. All of these indicate that the hybrid nanofilm is more suitable for constructing a drug delivery platform than PTB, because without the breaking or detachment of the coating on an implant surface, the corresponding leakage of the encapsulated drug and the related hazardous system exposure are not prone to occur.

## Antibacterial biofilm formation and biocompatibility evaluation

Urinary tract infection (UTI) is the main direct complication of catheterization and transurethral intervention, accounting for more than 30% of nosocomial infections[39]. According to reports, over 50% of catheters are colonized by bacteria within the first 10–14 days after being indwelled[40], and the biofilm composed of adherent bacteria and extracellular polymers secreted by them are particularly difficult to remove once formed[41]. By protecting bacteria from antibiotics and the body's immune system, biofilms often complicate UTI[42]. A previous study by our group showed that similar to electrically neutral zwitterionic polymers, PTB exhibits an almost balanced distribution of positive and negative amino acids on its surface and, therefore, can resist protein and bacterial contamination through neutralized surface hydration[22,43], making it a promising candidate for coating medical devices. Similar to the PTB, the PTB@SA nanofilm still exhibited a low negative surface zeta potential (Fig. 4a), so it may keep a good anti-biofilm ability, and this antibiotic-free antifouling toward microbial adhesion does not rely on the use of antibiotics, thereby would not induce bacterial drug resistance. As shown by colony counting and scanning electron microscope (SEM), compared with an unmodified substrate, the hybrid coating reduced the colonization of *Staphylococcus aureus* (*S. aureus*) and *Escherichia coli* (*E. coli*) by over 96 and 99%, respectively, and no statistical difference of effectiveness against bacterial colonization was observed between the PTB and PTB@SA coating groups (Fig. 4b, c, Supplementary Fig. 15). Such performance was further demonstrated by a micropatterned PTB@SA coating with 100 μm characteristic stripes (Supplementary Fig. 16 a, b). After exposing the micropatterned hybrid coating to the bacterial suspension, SYTO 9 staining revealed that almost no bacteria adhered to the PTB@SA coating area (Fig. 4d, e, Supplementary Fig. 16 c, d).

In contrast to typical existing approaches, the present hybrid nanofilm did not contain any synthetic reagents such as monomers, initiators, and cross-linkers, which thus showed good cytocompatibility and blood compatibility, as supported by the cytotoxicity test of primary rat urethral fibroblasts (Supplementary Fig. 17) and low hemolysis ratio (less than 5% as judging from ASTM F756-0834) (Supplementary Fig. 18). After indwelling the pristine or hybrid coating-modified urinary catheter in the urethra of adult male rabbits for 1 month, H&E staining confirmed that the urethral mucosa and underlying mucosal muscles in the hybrid nanofilm group were free of edema and necrosis, indicating friendly histocompatibility (Supplementary Fig. 19). In addition, encrustation and blockage are also thorny challenges in constructing functionalized urinary catheters. In vitro encrustation experiment indicated that the PTB@SA coating did not induce more severe encrustation in artificial urine, artificial urine in the presence of *Proteus mirabilis* (*P. mirabilis*), or human urine compared with unmodified catheters (Supplementary Fig. 20). On the premise of the proven good biocompatibility, we further evaluated the in vivo anti-biofilm performance of the coating in the rabbit urinary system. At the end of 7 days of the catheterization period, the surface of the pristine SR catheter was contaminated by the dense *E. coli* biofilm, while only a few bacteria were scattered on the hybrid nanofilm-coated catheter (Fig. 4f, g), exhibiting an in vivo anti-biofilm efficiency of over 87% (Fig. 4h). As a result, the number of urinary leukocytes in rabbits treated with nanofilm-modified catheters was also significantly lower than that of the control group (Fig. 4i). Macroscopically, the urethra

became hyperemic and edematous in the control group (Fig. 4j), with higher edema scores (Supplementary Fig. 21), suggesting the occurrence of urethritis. As shown in Fig. 4k, the infiltrated area of cells with a high nuclear-to-plasma ratio (probably inflammatory cells) in the control group was 10 times that of the coated group (Fig. 4k, l). Furthermore, immunochemical staining for CD 45 (expressed in leukocytes) and CD 68 (expressed in macrophages) showed that modification with the hybrid nanofilm resulted in a markedly imperceptible inflammatory response confined to the epithelial layer, whereas more inflammatory cells infiltrated both the mucosal and submucosa of the control group (Fig. 4k, m). These inflammatory cells have the potential to induce and mediate urethral stricture through various cytokines (such as TGF-β1, TNF-α, and IL-6, etc.)[44], and even lead to the failure of urethroplasty[20]. In these respects, this hybrid nanofilm with potent anti-biofilm properties is well suited for designing topical drug delivery systems mounted on urinary catheters to modulate tissue healing.

## Drug immobilization and controlled release by forming a release rate-controlling membrane on a solid drug reservoir

Considering safety concerns and consistent targeting of the therapeutic window, the topical application of the anti-fibrotic drug would benefit from the improved drug delivery system with a stable release profile. For this purpose, we designed a solid reservoir-type drug delivery system by exploiting the molecular diffusion control function of the nanofilm[23]. Drug release from such a solid reservoir system involves the dissolution of the drug particles into solvated molecules and subsequent concentration gradient-driven diffusion of the drug molecules across the voids among closely packed biopolymer nanoparticles inside the PTB@SA nanofilm[45] (Fig. 5a). To verify such a model, the nanofilm formed at the air–liquid interface was first transferred to a PET support with a hole (dia. 6 mm). Then, a solid rapamycin reservoir was applied to the air side of the membrane as a constant source of activity, while the other side of the membrane was directly contacting with fresh release medium. As a control, an equal amount of rapamycin molecules dissolved in the release medium was applied to the air side of the membrane as a source of nonconstant activity (Fig. 5b). With the direct diffusion of drugs without the pre-solvation step, the release of rapamycin from the nonconstant liquid reservoir followed a first-order profile[46] and reached equilibrium after 19 h (Fig. 5c). In contrast, drug release from the solid reservoir was retarded because it must be solvated into drug molecules prior to diffusion. As a result, only 27.3% of rapamycin permeated the membrane from the solid reservoir after 19 h, and the release profile after 10 h typically followed a zero-order release (Fig. 5d, Supplementary Fig. 22, Supplementary Table S1). A small deviation from zero-order release at the early stage (before 10 h) may be related to the release of a small amount of drug entering into the nanofilm during the deposition of solid drug reservoir[47]. These results validated the feasibility of the PTB@SA nanofilm for constructing a reservoir system with a constant activity source and its superiority in stably controlled drug release.

Encouraged by the above results, we further designed a sandwich-like structure to immobilize rapamycin inside the nanofilm to integrate anti-biofilm property and anti-fibrotic activity into conventional urinary catheters (Fig. 6a–c, Supplementary Fig. 23). First, the urinary catheter was modified with an ultra-thin PTB@SA nanofilm (about 10 nm) to improve the poor wettability of SR and thus favor an even surface deposition of drug particles in the next step (Supplementary Fig. 24). Then, the drug particles with a diameter of several microns were deposited on the priming coating layer through solvent evaporation to construct a solid reservoir model. Finally, a hybrid PTB@SA nanofilm was formed in situ around the solid drug reservoir to encapsulate the solid drug cores on the catheter and act as the release rate-controlling membrane (visualization by mixing FITC-labeled BSA in the phase transition

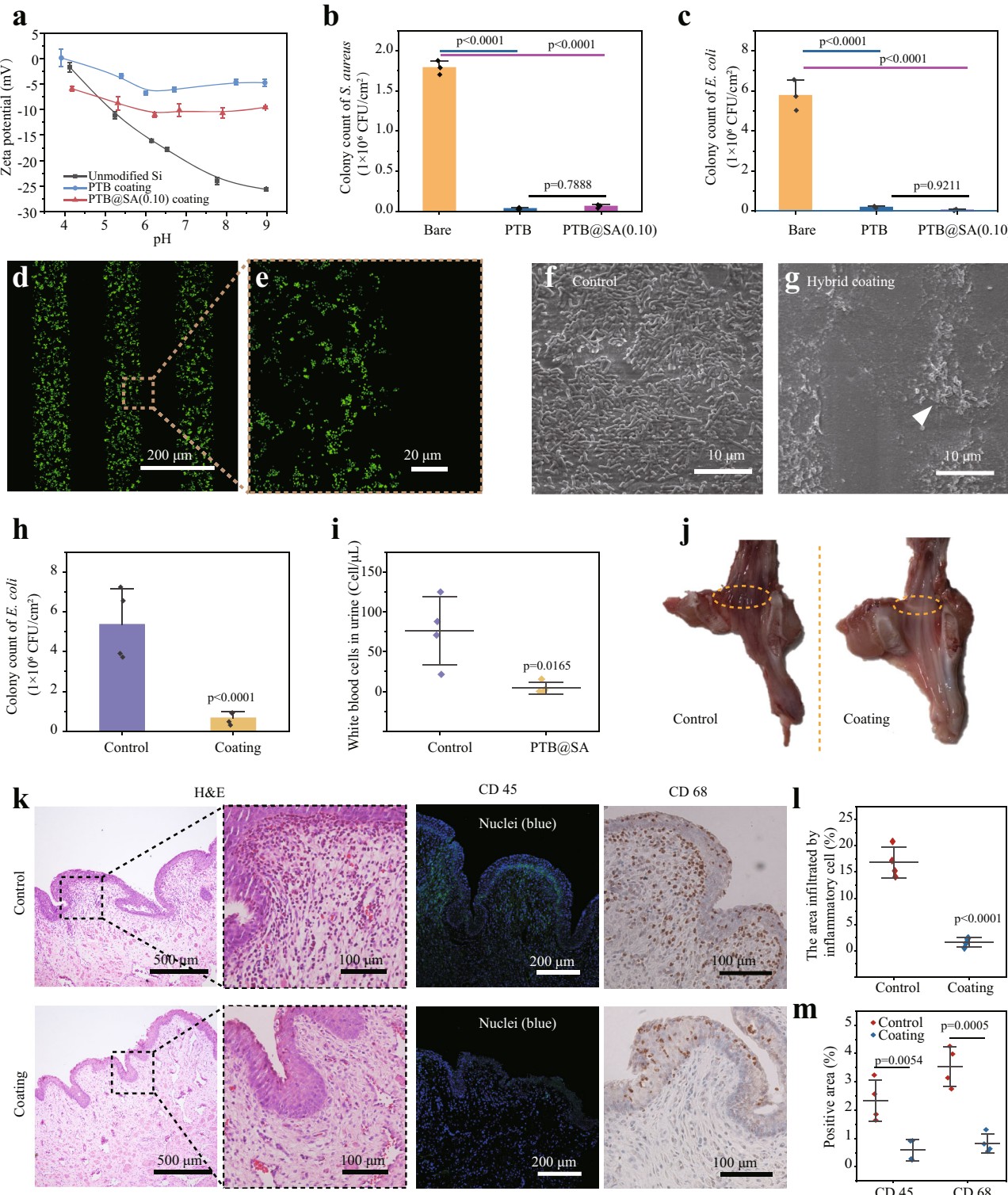

**Fig. 4 | The anti-biofilm performance and biocompatibility of the hybrid coating. a** Zeta potential of the PTB or PTB@SA(0.10) coating on Si. $n = 3$ independent samples per group. **b**, **c** Number of viable *S. aureus* (**b**) and *E. coli* (**c**) recovered from bare substrate, PTB, or PTB@SA(0.10) coating as confirmed by colony counting. $n = 3$ independent samples per group. **d**, **e** CLSM images showing the patterned adherence of *S. aureus* on a bare substrate using the micropatterned PTB@SA coating as the resistant layer. **f**, **g** SEM images of unmodified (**f**) or PTB@SA(0.10) coating-modified urinary catheters (**g**) recovered after indwelling for 1 week. The arrow indicates scattered bacteria on the PTB@SA(0.10) coating. **h** In vivo anti-biofilm efficiency of the hybrid coating measured by colony counting. $n = 4$ animals per group. **i** The number of white blood cells in rabbit urine after different catheters were used for 1 week. $n = 4$ animals per group. **j** Gross

observation of the rabbits' urethra and the yellow circles represent the sampling site of the tissue section. **k** Hematoxylin and eosin (H&E) staining and immunochemical staining with anti-CD 45 antibody (green) and anti-CD 68 (brown) antibody in rabbit urethral tissue. **l** Quantification of the area of inflammatory cell infiltration in the H&E staining images of the urethra. $n = 4$ animals per group. **m** Quantification of CD 45 (**l**) and CD 68 (**m**) positive area in Fig. 4k. $n = 4$ animals per group. The experiments in (**d**–**g**) were repeated independently at least three times with similar results. All data are mean ± S.D. Statistical significance was determined by one-way ANOVA with Tukey's multiple comparison test (**b**, **c**), two-tailed Student's *t*-test (**h**, **l**, **m**), or two-tailed Welch's *t*-tests (**i**). Source data are provided as a Source Data file.

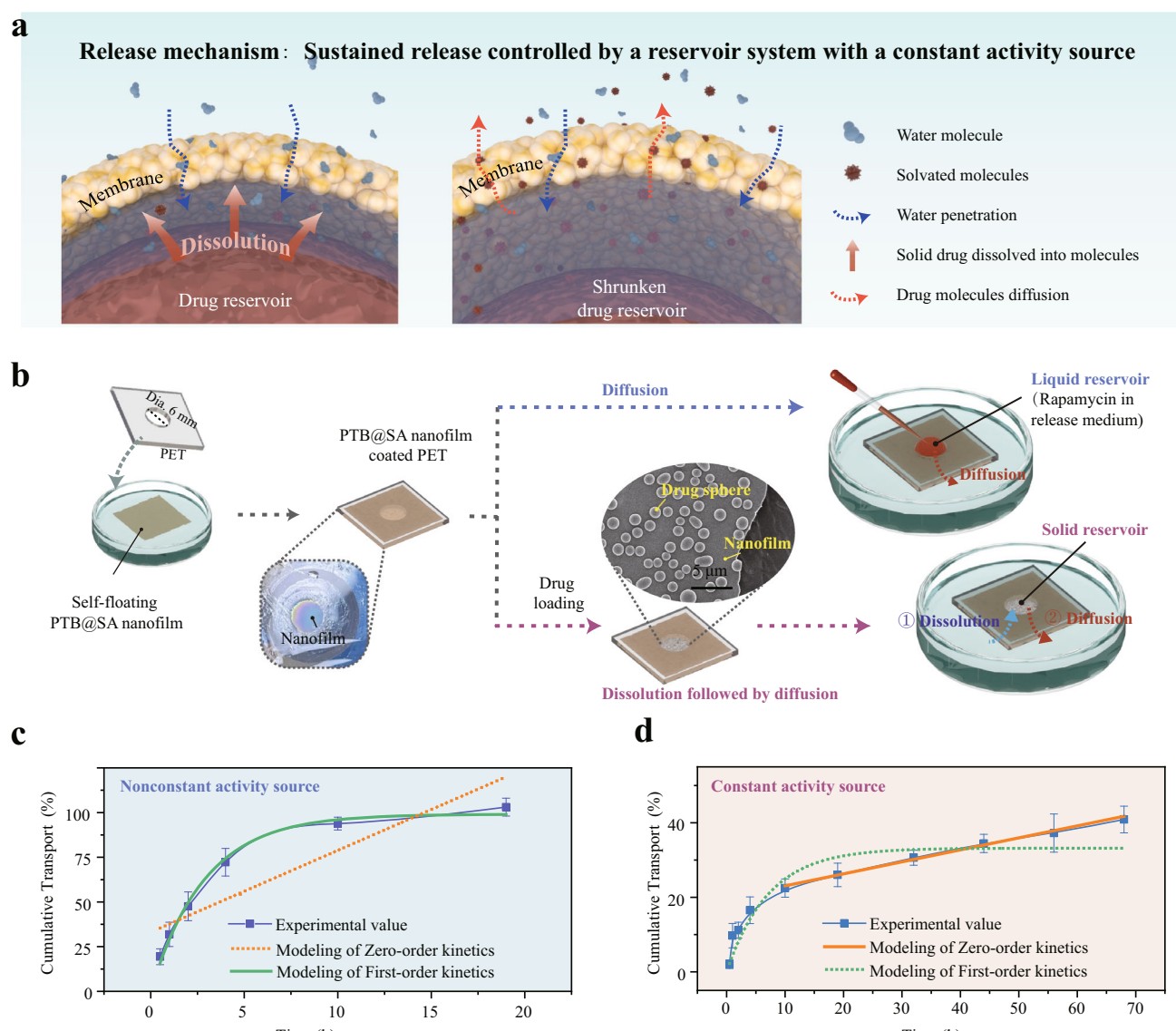

**Fig. 5 | Mechanisms of drug release from a reservoir-type system. a** Schematic showing the mechanism of drug release from a solid reservoir system consisting of a solid drug core and the surrounding membrane, including gradual dissolution into solvated molecules and subsequent diffusion through tortuous nanochannels among the biopolymer nanoparticles. **b** Illustration of the experimental setup for evaluating the release profiles presented by the constant (solid reservoir) or non-constant (liquid reservoir) activity source. **c, d** Experimental values and kinetic models for drug release through nanofilms from nonconstant (**c**) or constant activity sources (**d**). Data are mean ± S.D. $n$ = 5 independent samples per group. Source data are provided as a Source Data file.

system) (Fig. 6d, Supplementary Fig. 25). In such a way, the drug entrapment efficiency was determined as approximately 80% by acetonitrile elution, which was not largely affected by the content of SA in the release rate-controlling PTB@SA membrane (Supplementary Fig. 26). During the drug release, a continued depletion of the solid core reservoir could be clearly reflected by changes in the morphology of the delivery system tracked by SEM (Fig. 6e, Supplementary Fig. 27). With systematically exploring the effect of reaction temperature, final BSA concentration and SA addition ratio on the in vitro release behavior (Supplementary Figs. 28 and 29), it was further demonstrated that the PTB@SA nanofilm showed superior control capability to alleviate the initial burst release than the PTB nanofilm. Moreover, if native BSA or a mixture of native BSA and SA was only adsorbed non-specifically on the surface of drug particles, almost all drug content (more than 90%) was released within the first day. Although the PTB nanofilm extended the entire release period to 20 days, it still exhibited a pronounced burst release in the initial stage. In contrast, the alternative hybrid

PTB@SA nanofilm could reduce the amount of drug release in the first 24 h from 62.5% of the total drug loading (PTB) to a minimum of 15.6%. On the one hand, as the intermolecular hydrogen bonding between SA and PTB in the hybrid nanofilm reduced free hydrophilic groups and corresponding solvent uptake, the swelling ratio of the PTB@SA nanofilm was significantly lower than that of the PTB (Fig. 6f, Supplementary Fig. 30). As a result, incorporating SA in the release rate-controlling membrane retarded the step of solvent penetration through the membrane, as reflected by the reduced water permeability[48] (Fig. 6g, Supplementary Fig. 31). As such, the dissolution step of solid drug reservoir entrapped in the sandwich-like structure is noticeably restrained. On the other hand, substituting the PTB with the more flexible PTB@SA nanofilm also avoided undesirable drug leakage caused by the deformation-induced cracks (Fig. 6h, Supplementary Fig. 32). The release rate could be further derived from the first derivative of cumulative release with respect to time. The PTB@SA(0.10) nanofilm-controlled drug delivery entered a relatively stable release period

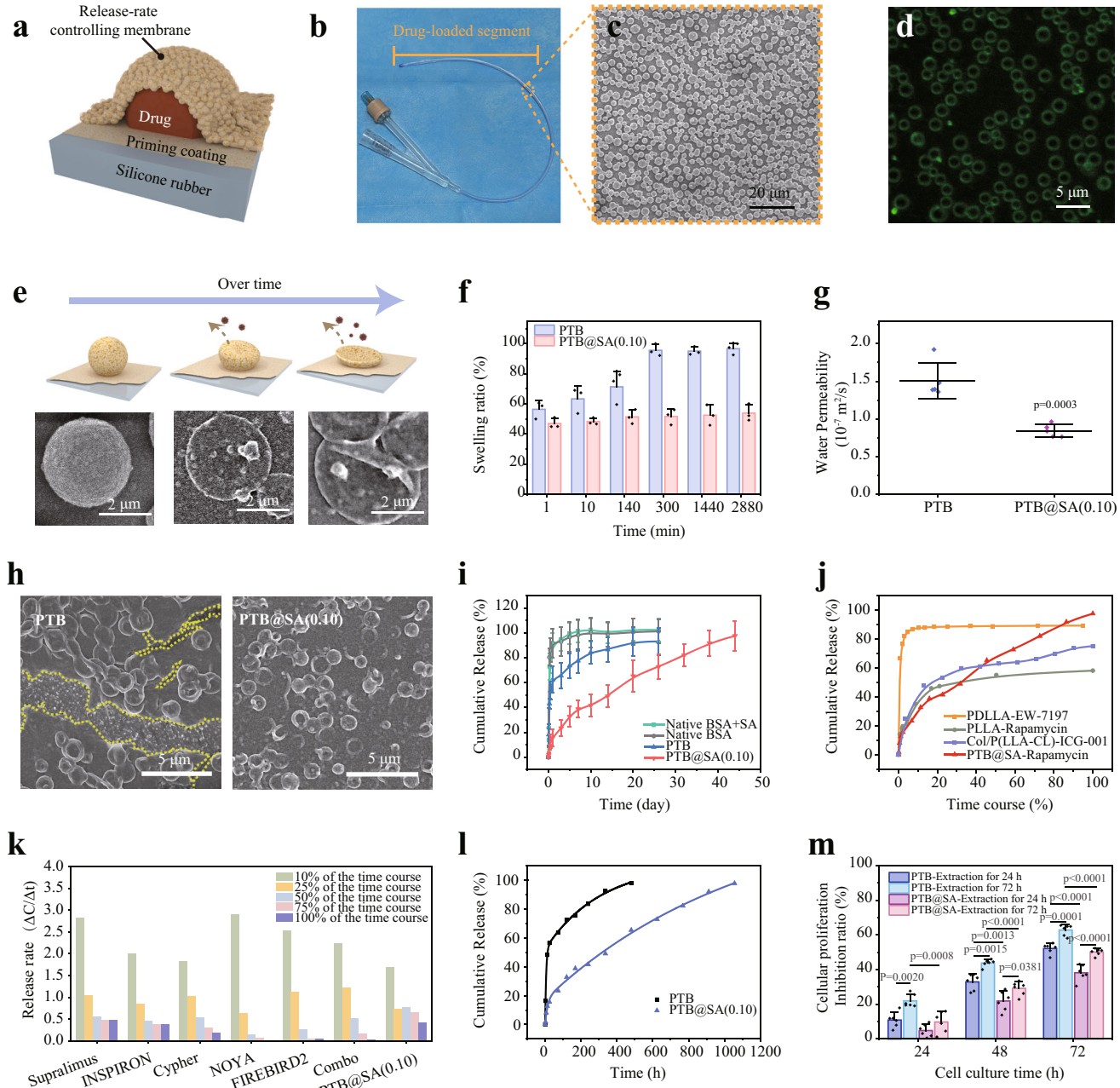

**Fig. 6 | The sandwich-like structure design to construct the release rate-controlling membrane-covered solid drug reservoir model on the medical implant. a** Schematic diagram showing the structure of the drug delivery system with a solid reservoir. **b, c** Representative optical (**b**) and SEM image (**c**) of the rapamycin sustained-release urinary catheter. **d** CLSM image of the PTB@SA nanofilm coated on the spherical drug particles by using FITC-labeled BSA to prepare the phase transition solution. **e** The morphology of the drug reservoir system at different release time points as tracked by SEM. **f** Swelling ratio of the PTB and PTB@SA(0.10) nanofilm in the release medium. $n = 3$ independent samples per group. **g** Water permeability of the PTB and PTB@SA(0.10) nanofilm. $n = 4$ independent samples per group. **h** SEM image of rapamycin delivery platform controlled by the PTB or PTB@SA(0.10) nanofilm on SR after releasing in the medium for 14 days. The yellow dotted lines represent the re-exposed SR after the PTB membrane ruptured. **i** Cumulative release of rapamycin from drug reservoirs controlled by different release rate-controlling materials. $n = 6$ independent samples per group. **j** The time-normalized release profiles of PTB@SA-Rapamycin and other topical delivery systems adapted from the literature for preventing urethral stricture. **k** Comparison of stable release capacity of commercially available rapamycin-eluting stents (Supplementary Table S3) and PTB@SA-Rapamycin. Release profiles of commercially available stents were adapted from literature, including time normalization and subsequent first-order derivation of cumulative percentage of drug release with respect to time. **l** Biphasic kinetic model was used to fit the release kinetics of rapamycin. **m** The proliferation inhibition ratio of primary rat urethral fibroblasts treated with different extracts as determined by 3-(4,5-Dimethylthiazol-2-yl)−2,5-diphenyltetrazolium bromide (MTT) assay. $n = 6$ independent cells per group. All data are mean ± S.D. Statistical significance was determined by two-tailed Student's $t$-test (**g**) or one-way ANOVA with Tukey's multiple comparison test (**m**). Source data are provided as a Source Data file.

after the first week. Until the 30th day, 1.52% of the total drug loading was still released from the PTB@SA(0.10) nanofilm per day, while the PTB system delivered less than 1% per day after 14 days (Fig. 6i, Supplementary Fig. 33).

Similarity analysis ($f_2$) was further performed to obtain quantitative values from the apparent differences in the release profiles[49]. Compared with the nonspecific adsorption of native BSA or the mixture of native BSA and SA on the reservoir, both PTB and

PTB@SA(0.10) showed similar factors of less than 50, confirming that the presence of the PTB or PTB@SA nanofilm does make the drug release profile different from those of the corresponding nonspecific adsorption groups. Furthermore, the $f_2$ value between PTB and PTB@SA was 34.5 (Supplementary Table S2), indicating that the presence of SA in the release rate-controlling membrane further significantly improved the drug release behavior. Whether compared with various commercially available rapamycin polymer stents[50–53] or other studies utilizing intraurethral drug delivery to inhibit urethral stricture[54,55,56], the rapamycin administration strategy controlled by the PTB@SA(0.10) nanofilm showed a well-proportioned release rate throughout the release process (Fig. 6j, k, Supplementary Fig. 34, Supplementary Table S3). Different dynamics models were then used to fit the release pattern (Fig. 6l, Supplementary Fig. 35, Supplementary Table S4). Although both fit with the biphasic kinetic model best (correlation coefficient $R^2 > 0.99$), it should be noted that the release rate constant of PTB@SA(0.10) was much lower than that of PTB, indicating that the release controlled by the hybrid nanofilm was more similar to the zero-order model. In another test, we cultured primary rat urethral fibroblasts and examined their proliferation with different extracts obtained by immersing the sustained-release coating in cell culture medium for 24 h or 72 h, respectively (Fig. 6m). Uncontrolled proliferation of fibroblasts and the resulting excess matrix (primarily collagen) are involved in the development of tissue fibrosis, making fibroblasts a potential target for the treatment of fibrotic diseases such as urethral strictures. Regarding this work, the antiproliferative effect is directly positively related to the amount of drug released from the sustained-release system, so the longer extraction time led to a stronger inhibitory effect. While at the same extraction time, the proliferation inhibition ratio of the PTB@SA group was significantly lower than that of the PTB system because the alternative hybrid nanofilm alleviated the initial burst release of drugs (decreasing the drug concentration in the extracts), which was consistent with the in vitro release characteristics. The above results confirmed the efficacy of this drug delivery system at the cellular level by inhibiting the proliferation of fibroblasts and its potential for further application in fibrotic diseases.

## Rapamycin-loaded urinary catheter inhibits urethral stricture by regulating ECM metabolism after injury

To evaluate the therapeutic effect and advantages of this local sustained-release rapamycin (PTB@SA-Rapamycin) catheter in inhibiting urethral stricture, rabbits subjected to electrocoagulation-induced urethral injury were divided into the following subgroups (Supplementary Fig. 36): control (no treatment), unmodified urinary catheters, systemic administration (rapamycin 1.5 mg/kg/d, orally), the PTB@SA nanofilm-modified urinary catheters (without drug loading), burst-releasing catheters (without the release rate-controlling membrane around the drug reservoir, but with the ultra-thin PTB@SA nanofilm as the priming layer on catheter before drug deposition), and PTB@SA-Rapamycin catheter. After 1 month, retrograde urethrography was performed to evaluate the configuration of the lumen, and the degree of urethral stricture was determined by measuring the diameter of the narrowest segment of the stricture and the distal urethra[57]. The obvious urethral stricture in the blank control group indicates that the animal model has been successfully established. In this case, the anti-fibrotic catheter (i.e., PTB@SA-Rapamycin catheter) could decrease the lumen reduction from 51.8% (unmodified catheter) to 10.8% (Fig. 7a, b), making the diameter of the repaired urethra close to that of normal healthy rabbits (Supplementary Fig. 37). Such a high mean percent improvement in lumen stenosis of 79.2% suggests a potentially great advantage of the PTB@SA-rapamycin catheter in preventing urethral strictures compared with other topical delivery regimens (Fig. 7c, Supplementary Table S5). The mucosa of the healed urethra exposed to the anti-fibrotic catheter was smooth and ruddy,

and the abundance of submucosal fibrous connective tissue was almost identical to the adjacent normal tissue, while whitish, stiff, wrinkled scar tissue was visible in the remaining groups, showing significant submucosal fibrosis histologically (Fig. 7d, Supplementary Figs. 38 and 39). Immunochemical staining indicated that α-smooth muscle actin (α-SMA, an indication for trans-differentiation of fibroblasts into myofibroblasts[7]) had the lowest expression level in the PTB@SA-Rapamycin catheter group, while the expression of matrix metalloproteinase 1 (MMP1) increased simultaneously. It can thus be deduced that the PTB@SA-Rapamycin catheter ameliorated ECM disorder by reducing fibroblast-derived collagen production and enhancing MMP1-induced collagen degradation, reducing submucosal collagen to a level similar to the physiological condition (Fig. 7d, e, Supplementary Fig. 40). In contrast, limited by poor bioavailability and insufficient duration[58], neither systemic administration nor burst-releasing catheter cannot inhibit urethral stricture formation (Fig. 7a, b). At 2 weeks after the catheter removal, the urethral diameter of rabbits in the PTB@SA-Rapamycin catheter group was still significantly wider than that in the unmodified catheter group. H&E and Masson staining also consistently highlighted the role of PTB@SA-Rapamycin catheters in inhibiting fibrosis progression after urethral injury, and lower SMA expression in the anti-fibrotic catheter group predicted a better prognosis than the unmodified catheter group (Supplementary Fig. 41)[59,60]. The above results suggest that the anti-fibrotic catheter has shown impressive preliminary results in retarding the progression of urethral injuries to strictures, but does have a limitation that its long-term effect (6 months to 1 year) in inhibiting urethral strictures need to be clarified in future experiments.

Controllable sustained release of rapamycin from the PTB@SA-Rapamycin catheter further showed a low systemic spillover. By collecting whole blood at predetermined time points and analyzing the corresponding rapamycin concentrations, it was then determined that the systemic exposure induced by the PTB@SA-Rapamycin catheter during the first 24 h was much lower than that of the burst-releasing group, illustrating the effectiveness of the release rate-controlling membrane (PTB@SA) in decelerating the drug release in vivo (Supplementary Fig. 42). In addition, with the release rate of the anti-fibrotic catheter reaching a stable state after 1 week, the whole blood concentration of rapamycin also fell below the detection limit of 0.2 µg/L, whereas oral administration induced an enhanced systemic exposure (up to over 25 ng/mL). Long-term or high-dose systemic exposure caused by oral administration is clearly associated with the occurrence of drug side effects, while in contrast, the limited systemic spillover from the PTB@SA-Rapamycin catheter ensures good biosafety, as reflected by histological analysis of major organs (heart, liver, spleen, lung, kidney, stomach, intestine, bladder) and changes in liver and kidney function. Compared with the normal healthy group, the PTB@SA-Rapamycin catheter did not cause obvious tissue damage or pathological changes of major organs after indwelling for 30 days (Supplementary Fig. 43), and blood biochemical indexes did not change significantly compared with those before treatment. In contrast, alanine aminotransferase (ALT) and aspartate aminotransferase (AST) increased to twice the initial levels in the systemic administration group (Fig. 7f, g, Supplementary Fig. 44), and corresponding liver tissue staining indicated hepatocyte ballooning and hepatocyte injury (Fig. 7h), an adverse event associated with systemic administration of rapamycin[61,62]. The body weight of the rabbits in each group showed a consistent slight drop within 2 weeks after urethral coagulation and then gradually recovered, which may be related to surgical stress (Supplementary Fig. 45). These results indicated that the anti-fibrotic PTB@SA-Rapamycin urinary catheter with an increased effect/dose ratio (accounting 2.2% and 10.7% of the dose required for systemic administration or urethral irrigation, respectively) could significantly limit systemic toxicity and safely target the urethra by maintaining normal ECM homeostasis.

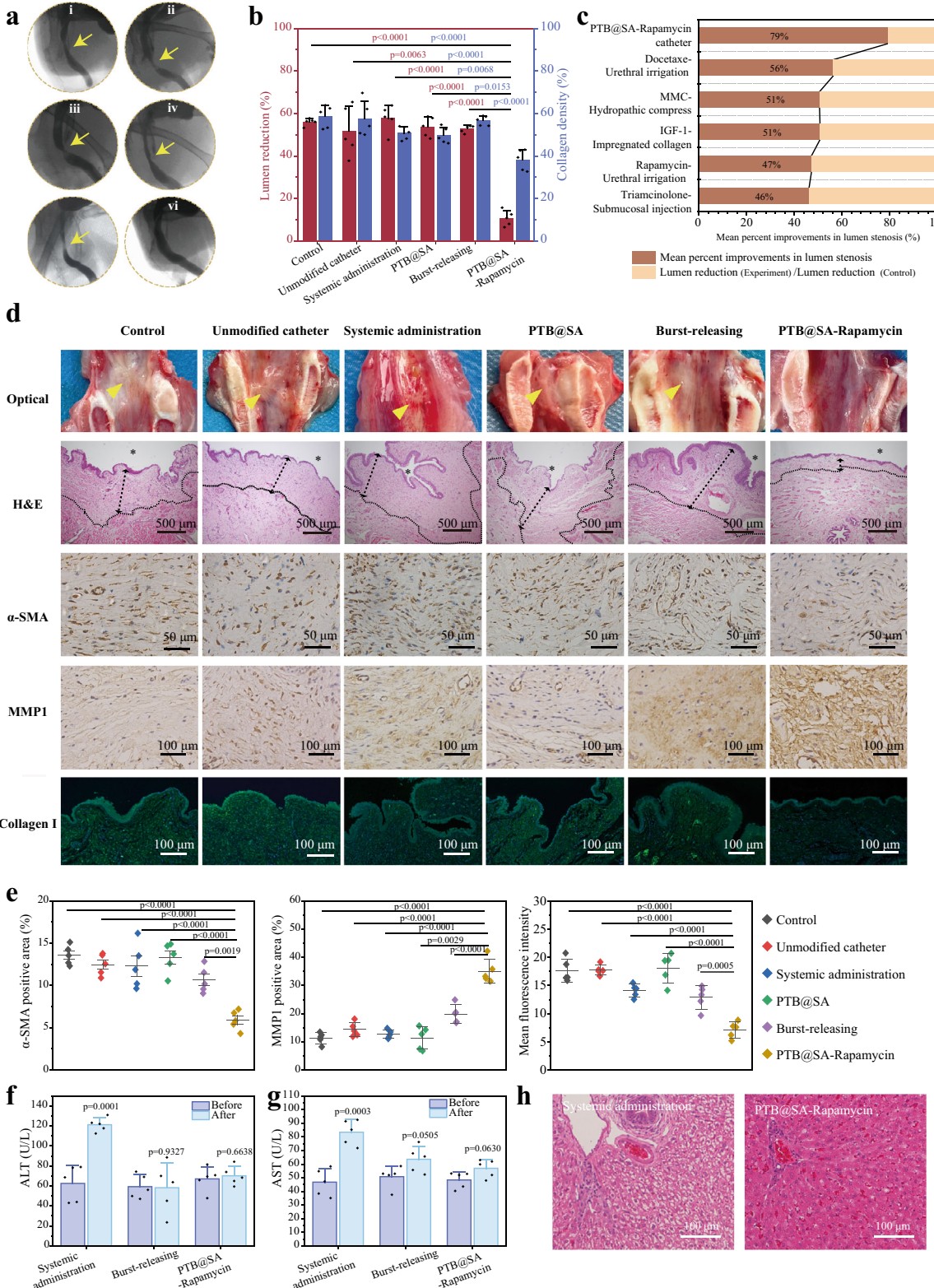

In summary, anti-fibrotic catheters using polysaccharide-protein hybrid nanofilm assembled from biopolymer nanoparticles at the interface provides a considerable therapeutic benefit in preventing urethral stricture and catheter-related urinary tract infections. In this hybrid nanofilm, significantly improved toughness resulting from the integration of SA and supramolecular interaction between SA and unfolded protein chains has been finely coupled with multiple other favorable properties, typically including robust adhesion on various surfaces, excellent anti-biofilm properties and nanochannels intrinsically existing in the as-formed hybrid nanofilm. As a result, this hybrid nanofilm is considered a promising candidate for constructing sustained-release drug delivery coating on medical materials. Rapamycin is a therapeutic agent with anti-fibrotic effects, and its beneficial effect in intervening in the fibrosis process after urethral injury at the

**Fig. 7 | The sustained-release coating of rapamycin on the urinary catheter attenuates collagen deposition and inhibits the formation of strictures post urethral injury. a** Representative retrograde urethrogram of rabbits with different treatments after urethral electrocoagulation, using 76% Meglumine Diatrizoate as a contrast agent. **i** Control; **ii** Unmodified catheter; **iii** Systemic administration; **iv** PTB@SA; **v** Burst-releasing; **vi** PTB@SA-Rapamycin. **b** Quantitative analysis of urethral lumen reduction and submucosal collagen density in rabbits with different treatments. $n = 5$ animals per group. Statistical significance for lumen reduction was determined by one-way ANOVA with Dunnett's multiple comparison test, whereas statistical significance for collagen density was determined by one-way ANOVA with Tukey's multiple comparison test. **c** Corresponding results of the PTB@SA-Rapamycin catheter compared with other preclinical studies of topical therapy to suppress urethral stricture. **d** Gross specimen, H&E staining and immunohistochemical staining with anti-alpha smooth muscle actin antibody (α-SMA, brown), anti-matrix

metalloproteinase 1 antibody (MMP1, brown), and type I collagen antibody (Collagen I, green) of the urethral injury sites at 1 month. Nuclei (blue) were stained with DAPI. Yellow triangles indicate urethral scar tissue. The asterisks represent the urethral luminal side, and the dotted lines indicate the boundary between the submucosa and the muscularis. **e** Quantification of α-SMA, MMP1 positive area, and Collagen I fluorescence intensity in Fig. 7d. $n = 5$ animals per group. Statistical significance was determined by one-way ANOVA with Tukey's multiple comparison test. **f, g** Hematological examination to show the changes of alanine aminotransferase (ALT, **f**) and aspartate aminotransferase (AST, **g**) with different treatments for 1 month. $n = 5$ animals per group. Statistical significance was determined by two-tailed Student's $t$-test. **h** H&E staining of rabbit liver after systemic administration or indwelling rapamycin sustained-release catheter for 1 month. All data are mean ± S.D. Source data are provided as a Source Data file.

cellular and animal levels has been confirmed in our previous studies[14,63]. In addition, rapamycin-eluting coronary stents have shown reliable efficacy and negligible side effects in long-term clinical practice. These all encourage us to utilize the hybrid nanofilm to immobilize solid drug reservoirs, forming a sandwich-like solid reservoir system with a constant activity source to stably release anti-fibrotic agents for at least tens of days. Consequently, conventional urinary catheters are tactfully imparted with both anti-fibrotic and anti-biofilm properties in a one-step preparation process. By targeting the relevant pathological microenvironment, the sustained-release catheter requires only 10.7% of the dose demanded urethral irrigation and reduces lumen stenosis by 79.2%, ranking it at the top level among the preclinical studies on local therapy to prevent urethral stricture. In this regard, the promotion of PTB@SA-Rapamycin catheters as an alternative to conventional catheters is expected to be of great benefit to numerous people at high risk for urethral strictures (eg, undergoing transurethral intervention, traumatic catheterization, or urethroplasty). Our study blazes a trail in the management of benign luminal stenosis characterized by ECM metabolic disorders. This study also proposes a scale-up strategy for implant surface functionalization based on facile protein nanofilm coating, robust interfacial binding, and controlled molecular release. In this regard, it is reasonably envisaged that the present concept is not limited to specific proteins, catheters, and drugs and could generally be extendable to other proteinaceous nanofilms, medical implants, and important medicinal molecules. For instance, besides rapamycin and BSA used as model drugs and proteins in the present work, this strategy is applicable to the long-term delivery of other drugs such as paclitaxel (Supplementary Fig. 46), and human serum albumin (HSA) is also expected to form a drug-releasing coating (Supplementary Fig. 47), suggesting the versatility and generality of this drug delivery system. This approach thereby may greatly motivate the further marriage between the realms of nanotechnology and biointerface.

## Methods
### Materials
Bovine serum albumin (BSA), polyoxyethylene (10) cetyl ether (Brij 58), rhodamine, Thioflavin T (ThT), rapamycin, thiazolyl blue tetrazolium bromide, and paraformaldehyde were purchased from Sigma-Aldrich. SA was purchased from Aladdin. Paclitaxel was purchased from Macklin. Phosphate buffered saline (PBS), Mueller–Hinton broth (MHB), Mueller–Hinton agar (MHA), and methyl blue (MB) were purchased from Solarbio. Tris (2-carboxyethyl) phosphine hydrochloride (TCEP) was purchased from TCI. Dimethyl sulfoxide (DMSO), ethanol, acetonitrile, 30% hydrogen peroxide ($H_2O_2$), sodium hydroxide (NaOH), sulfuric acid ($H_2SO_4$), and Tween 80 were purchased from Sinopharm Chemical Reagent Co., Ltd. 5(6)-aminofluorescein-labeled SA was purchased from Ruixi Biological Technology Co., Ltd. *Staphylococcus aureus* (*S. aureus*) (ATCC6538) and *Escherichia coli* (*E. coli*) (ATCC8739) were obtained from American Type Culture Collection

(USA). The silicone urinary catheter was purchased from Wellead Medical Instruments Co., Ltd. (Guangzhou, China).

### Preparation of the hybrid PTB@SA nanofilm
The phase transition solution was prepared by premixing BSA and SA in a mass ratio of 1:0.05–0.25, followed by the addition of an equal volume of TCEP aqueous solution (50 mM at pH 4.5, pH adjusted by 5 M NaOH). To assemble the nanofilm at the solid–liquid interface, the clean substrate was completely immersed in the freshly prepared phase transition solution and placed in a humid environment. The coated substrate was then rinsed with ultrapure water to wash away unreacted BSA, SA, and other salts adsorbed on the surface. To assemble the nanofilm at the air–liquid interface, the protein phase transition solution was dropped on a piece of glass (e.g., 20 × 20 cm). Then, the solution on the substrate was incubated in a humid environment at 30 °C for 3 h. After the reaction was complete, the glass holding the reacted phase transition solution was transferred to the water to obtain the nanofilm floating at the air–liquid interface.

### ThT staining
After mixing BSA, water, and SA in a volume ratio of 1:0.3:0.2, 10 μL of 1 mM ThT solution was pipetted into 100 μL of the above-mixed solution, and then 66.6 μL of TCEP solution was added to the system. The fluorescence intensity of the solution was then measured by a multi-mode microplate reader (BioTek NEO2, BioTek, Winooski, VT) with excitation at 440 nm and emission at 484 nm. The reaction temperature and detection interval were set at 30 °C and 5 min, respectively.

### Determination of the encapsulation density of SA in the coating
To determine the actual content of the polysaccharide encapsulated in the coating, 5(6)-aminofluorescein-labeled SA was added to the phase transition solution. For this aim, the hybrid coatings prepared in the dark with different SA feed ratios were treated with 2 M NaOH for 4 h to dissolve the coating completely[64]. Therefore, the encapsulation density of SA in the coating was calculated by measuring the fluorescence intensity in NaOH.

### QCM-D
The QCM-D measurements of native BSA, the phase-transitioned BSA (PTB), and the PTB doped with SA (PTB@SA) on Au were carried out on a Q-sense E1 instrument (Biolin, Sweden). The changes in resonance frequency (ΔF) and dissipation (ΔD) were recorded at the fundamental frequency (4.95 MHz) and its 3rd, 5th, 7th, 9th, and 11th overtones. For the adsorption of each sample, a clean Au chip was inserted into the flow chamber and incubated in Milli-Q water at a flow rate of 200 μL/min. After baseline equilibration, the solution to be tested was pumped into the flow chamber. After 1 h, the flow chamber was alternatively rinsed with Milli-Q water. The whole measurement was performed at 25 °C, and the flow rate was 200 μL/min. The

adsorption amount was calculated by QSense Dfind (version 1.2.1) in the Dfind Smartfit model.

## Coating stability evaluation by thickness test

The hybrid PTB@SA(0.10) coating on the silicon wafer with an original thickness of 50 nm was subjected to the following treatments, including immersion in ethanol or dimethyl sulfoxide (DMSO) for 4 h, ultrasonic for 10 min, 3 M adhesive tape peeling for three times, autoclave sterilization at 121 °C for 21 min, and immersion in 2% Tween 80, 0.5% Brij 58-PBS or artificial urine for 30 days. After these treatments, the coating thickness was re-evaluated.

## Uniaxial tensile test of the nanofilm

The film samples were collected by multilayer stacking of the nanofilms formed at the air–liquid interface and uniaxially stretched with a tensile machine. The force–displacement curve was recorded in the loading mode with a constant displacement speed of 1 mm/min. Toughness or energy absorption capacity was determined from the area under the stress–strain curve.

## Nanoindentation measurements

For nanoindentation force analysis, a multilayer stacking sample of the nanofilms with a thickness of 4 μm was fabricated on a silicon wafer. The mechanical property of the PTB or hybrid nanofilm was then measured using a nanoindenter apparatus (TI 950 Tribo-Indenter; CSM, Switzerland) equipped with a diamond Berkovich tip. Load-controlled tests at ±200 μN/s to peak force (1000 μN) were performed with 2 s holding time.

## Fabrication of micropatterned coatings

To obtain patterned coatings, positive patterning on the hybrid coating was performed by exposing the nanofilm coating to the UV light at 8000 μw/cm² (mainly 254 nm wavelength) from the topside (a high-pressure mercury lamp, 1000 W) for 10 min. In this process, the coating-modified substrate was held between a quartz plate (on the bottom) and a photomask with striped chrome patterns (on the top). After exposure to UV light, these samples were rinsed in ultrapure water for 1 min. Coatings with strip-shaped micropatterns with a pitch of 100 μm can be obtained by the above method.

## Anti-biofilm formation performance

The bacteria in the logarithmic growth phase were centrifuged and suspended in fresh MHB, then the bacterial suspension of 300 μL with an initial concentration of $10^6$ CFU/mL was incubated with the bare substrate, PTB, or hybrid nanofilm-coated substrates in a humid chamber. After 24 h incubation at 37 °C, the sample was washed with PBS to remove the floating bacteria, and then the sample was placed in a centrifuge tube containing 10 mL of PBS and sonicated at 200 W/ 40 HZ for 5 min to resuspend the bacteria adhered to the surface in PBS. The bacterial suspension was serially diluted and plated on MHA plates. After the bacteria grow to be visible to the naked eye, the number of clones was counted to evaluate the bacterial load in the biofilm on the material. For samples observed by SEM, after incubation with the bacteria, all samples were washed with PBS and then fixed in 4% paraformaldehyde PBS buffer overnight.

For samples stained with SYTO 9, after rinsing to remove planktonic bacteria, the samples were stained by immersing in SYTO 9 solution (100 nM) for 10 min and then rinsed with normal saline.

## Drug release from nanofilm-controlled reservoir-type delivery systems

The PTB@SA(0.10) hybrid nanofilm formed by the phase transition solution at the gas–liquid interface was floated in ultrapure water and subsequently transferred to a PET support with a hole (6 mm). One side of the nanofilm was directly contacted with 5 mL of 0.5% Brij 58-PBS to form a receiving pool, and the other side was applied with 50 μL of 5 mg/mL rapamycin dissolved in 0.5% Brij 58-PBS solution or an equivalent amount of solid drug reservoir to form a nonconstant activity source (liquid reservoir) or a constant activity source, respectively. At the indicated time points, 0.5 mL of the underlying release medium was pipetted, and the rapamycin concentration in it was determined by the UV spectrophotometer (UV5, Mettler Toledo, Greifensee, Switzerland), and the cumulative percentage of rapamycin that penetrated the nanofilm was calculated.

## Manufacture of anti-fibrotic catheters

The cleaned SR catheters were immersed in a phase transition solution with final concentrations of BSA, SA, and TCEP (pH 4.5) of 0.8 mg/mL, 0.08 mg/mL, and 20 mM, respectively. After incubation at 25 °C for 2 h, the samples were washed with ultrapure water to remove unreacted salts and proteins. In this way, the catheter surface was successfully modified with a priming coating. Rapamycin dissolved in ethanol was dropped on the priming coating at a loading density of 300 μg/cm², and the drug reservoir was deposited after the solvent evaporated. Next, the drug-loaded catheters were re-soaked in a phase transition solution with final concentrations of BSA and TCEP of 8 mg/mL and 20 mM, respectively, and final concentrations of SA ranging from 0.8 to 2 mg/mL. After incubation at 30 °C for 3 h, a release rate-controlling membrane with different SA doping amounts was formed in situ on the surface of the drug reservoir. The anti-fibrotic catheter can be obtained after the above-mentioned samples are washed with ultrapure water and dried at room temperature. For the PTB nanofilm-controlled drug delivery system, SA in the phase transition solution used to prepare the priming coating and release rate-controlling membrane was replaced with an equal volume of water. For catheters with nonspecific adsorption of BSA or a mixture of BSA and SA on the surface of the drug reservoir, the TCEP in the phase transition solution was replaced by an equal volume of ultrapure water.

## Establishment of primary rat urethra fibroblasts

After the rat was sacrificed, its foreskin was incised, and the urethra was bluntly separated. Then, the fresh urethra tissue was rinsed with Hanks Balanced Salt Solution (Solarbio) to remove residue plasma. Next, the tissue was cut into small pieces (1 mm³) and treated with collagenase (0.2%, Sigma-Aldrich, St. Louis, MO) at 37 °C for 40 min, and trypsin (Sigma- Aldrich, St. Louis, MO) was added in the last 5 min. These small pieces of tissues were put in a culture flask with Dulbecco's modified Eagle's medium (DMEM; Invitrogen, Carlsbad, CA) and supplemented with 10% of fetal bovine serum (Biological Industries, Kibbutz Beit Haemek, Israel) and 1% penicillin-streptomycin (Sigma-Aldrich, St. Louis, MO) in a humidified atmosphere at 37 °C with 5% CO₂. After cells reached 70% of confluence, the explants were removed, and the cells grew into the remaining spaces. After reaching 90% confluence, cells were detached with trypsin for 5 min at 37 °C. Primary rat urethral fibroblasts within 3 to 7 passages were used in this study.

## In vitro release

The rapamycin sustained-release urinary catheter (PTB@SA-Rapamycin) was cut into segments with a length of 1 cm and then placed in a shaker at 37 °C with a rotating speed of 70 r/min. At the indicated times, 0.5 mL of the release medium was withdrawn, and the ultraviolet absorbance was measured by a UV-spectrometer (UV5, Mettler Toledo, Greifensee, Switzerland), while the same volume of fresh medium was added to each sample.

The release rate was calculated as follows:

$$\text{Release rate} = \frac{\Delta C}{\Delta t} \times 100\% = \frac{C_{n+1} - C_n}{t_{n+1} - t_n} \times 100\% \qquad (1)$$

$\Delta C$ represents the difference between the cumulative percentage of drug release at adjacent time point, and $\Delta t$ represents the corresponding time interval. $C_{n+1}$ and $C_n$ represent the cumulative percentage of drug release calculated from the release medium removed for the $n^{\text{th}}$ or $(n+1)^{\text{th}}$ time, while $t_{n+1}$ and $t_n$ represents the corresponding time point.

Similarity factor ($f_2$) analysis was used to compare in vitro release profiles.

$$f_2 = 50 \times \text{Log}\left\{\left[1 + \frac{1}{n}\sum_{t=1}^{n}(R_t - T_t)^2\right]^{-0.5} \times 100\right\} \qquad (2)$$

Where $n$ is the number of test points, and $R_t$ and $T_t$ are the percentage of drug released at each time point for the two compared formulations, respectively. The US Food and Drug Administration and the European Agency for the Evaluation of Medicines recommend that two release profiles can be considered similar if $f_2$ is between 50 and 100[65].

### In vitro drug release kinetics

The drug release patterns of rapamycin controlled by the PTB or PTB@SA(0.10) nanofilms were analyzed by zero-order, first-order, Higuchi, and biexponential and biphasic kinetics models, respectively. Mathematical equations of the kinetic models employed are as follows:

$$\text{Zero} - \text{order} : Q = kt + C \qquad (3)$$

$$\text{First} - \text{order} : \ln(1 - Q) = -kt \qquad (4)$$

$$\text{Higuchi} : Q = kt^{0.5} + C \qquad (5)$$

$$\text{Biexponential and biphasic kinetics} : Q_0 - Q = Ae^{\alpha t} + Be^{\beta t} \qquad (6)$$

Where $Q$ represents the cumulative percentage of drug release at time $t$; k was a kinetic constant measuring the release rate; C is a constant. The biexponential and biphasic kinetic model consists of fast (burst release) and slow (sustained release) phases. In this equation, α and β are the release rate constants of burst release and sustained release, respectively. A and B are the kinetic constants of burst release and sustained release, respectively[66].

### The proliferation inhibition of primary rat urethral fibroblasts by the sustained-release coating

Rapamycin was immobilized at a density of 300 μg/cm$^2$ to obtain a sustained-release coating controlled by the PTB or PTB@SA(0.10) nanofilms. The 12 cm$^2$ sustained-release coating was then immersed in 2 mL of DMEM containing 10% FBS and extracted at 37 °C in a humidified atmosphere containing 5% CO$_2$ for 24 or 72 h to obtain the extract. Primary rat urethral fibroblasts in complete medium (100 μL) of 3000 cells in each well were seeded into a 96-well plate. After cell attachment, the complete medium in each well was replaced with 100 μL of extract. The 3-(4,5-Dimethylthiazol-2-yl)–2,5-diphenyltetrazolium bromide (MTT) assay was then performed to evaluate the inhibitory effect of rapamycin released from the release rate-controlling membrane with different components on the proliferation of primary urethral fibroblasts. The proliferation inhibition ratio is calculated according to the following formula:

$$\text{Proliferation inhibition ratio} = 100\% - \frac{OD(sample) - OD(blank)}{OD(control) - OD(blank)} \times 100\% \qquad (7)$$

Where OD (sample) is the optical density of the experimental group at a wavelength of 490 nm, OD (control) is the optical density of cells cultured with normal culture medium at a wavelength of 490 nm, and OD (blank) represents the optical density of the medium without cells.

### Animal experiment

All experiments were performed according to the guidelines of the Biomedical Ethics Committee of Health Science Center of Xi'an Jiaotong University (No. 2020-864). All animals were adult male New Zealand rabbits with a weight of 3.52 ± 0.54 kg. The evaluation of the antibacterial biofilm performance of the coating in the rabbit urethral system was carried out in the following steps. First, the pristine SR catheters and the PTB@SA(0.10) nanofilm-coated catheters were exposed to *E. coli* (ATCC8739) with the same bacterial concentration in artificial urine for 20 h at 37 °C to simulate possible microbial contamination. Eight New Zealand white rabbits were randomly divided into two groups and anesthetized with pentobarbital (30 mg/kg). The two groups of urinary catheters were flushed with PBS and then immediately placed in the urinary tract of New Zealand white rabbits. The rabbits were sacrificed after 1 week, and the urethral tissue and urine in the bladder were collected for tissue staining and routine urinalysis, respectively. Edema was scored semiquantitatively on a 3-point scale in tissue sections, with 0 being the lowest and 2 being the highest scores[67]. Recovered catheters were evaluated for bacterial density on the surface by SEM and colony counting, respectively. For in vivo experiment to evaluate the effect of inhibiting urethral stricture, after being anesthetized by sodium pentobarbital (30 mg/kg), rabbits were placed in a supine position, and a standard 10-mm-long circumferential electrocoagulation of the urethra outside the seminal orifice was induced through a 13-French pediatric resectoscope under aseptic conditions. The urethra at the same position in each rabbit was continuously electrocoagulated until the mucosa was bleached and ulcerated. During this process, the power of the hook-shaped electrode was set at 50 W. All electrocoagulation procedures were performed by the same urologist to ensure that all animals suffered a consistent degree and depth of urethral injury. After electrocoagulation, all animals were randomly divided into six groups with five animals in each group: (i) control: without any intervention; (ii) indwelling unmodified silicone catheter; (iii) systemic administration: feeding rapamycin at a dose of 1.5 mg/kg per day; (iv) PTB@SA: indwelling urinary catheter coated with PTB@SA(0.10) nanofilm without rapamycin; (v) burst-releasing: no release rate-controlling membrane on the surface of the drug particles; (vi) PTB@SA-Rapamycin: PTB@SA(0.10) nanofilm used as the release rate-controlling membrane for continuous local delivery. The catheters used above were all 6-French silicone catheters. The tip of the burst-releasing and sustained-release catheters was loaded with 3 mg rapamycin at a density of 600 μg/cm$^2$. Subsequently, the whole blood of rabbits in the systemic administration group, the burst-releasing and sustained-releasing group were collected at 2, 12, 24, 72 h, and 7, 14, 21, and 28 days, respectively. The concentration of rapamycin was determined by liquid chromatography-mass spectrometry (LC-MS). Retrograde urography was performed to observe the urethral gross morphology, and 76% of diatrizoate meglumine was slowly injected into the urethra as a contrast agent. After the rabbits were sacrificed, the urethra and major organs were collected and fixed in a 4% paraformaldehyde solution.

In order to exclude the interference of various experimental conditions to more objectively compare the preventive effect on urethral stricture reported in independent studies, we adopted the mean percent improvement in lumen stenosis for each intervention according to the following formula to quantify the improvement in the

severity of urethral strictures with different treatments.

$$\text{Mean percent improvement} =$$

$$\frac{\text{Lumen reduction(Control)} - \text{Lumen reduction(Experiment)}}{\text{Lumen reduction(Control)}} \times 100\%$$

$$(8)$$

## Characterizations

Far-UV CD (Applied Photophysics Ltd., England) spectra were collected under constant nitrogen flush at 25 °C and recorded from 190 to 260 nm with a 2.0 nm bandwidth, and Chiascan Spectrometer Control Panel Application (version 4.1.5) was used to process circular dichroism spectrum. FTIR spectra were obtained between 400 and 4000 $cm^{-1}$ with a resolution of 1 $cm^{-1}$ using an Alpha-T spectrometer (Bruker Tensor 27, Germany). A free-standing nanofilm directly adhered to the mold, and the freeze-dried powder of SA or HA was tableted with KBr for testing. Peak Fit (version 4) was used to analyze Fourier transform infrared (FTIR) spectra. After staining with ThT (0.1 mg/ml) for 1 min, a confocal laser scanning microscope (Olympus FV1200, Japan) was used to observe the 3D image of the hybrid nanofilm coated on a medical catheter with a complicated shape, and the corresponding data were collected by the FV1000 Viewer (version 3.1). XPS of the modified substrates was performed with an AXIS ULTRA instrument from Kratos Analytical Ltd., and the binding energies were calibrated by setting the $C1s$ peak at 284.6 eV. CasaXPS (version 2.3.16) was used to analyze X-ray photoelectron spectroscopy. The WCA of bare substrates and PTB@SA coated surfaces were determined on an OCA 20 instrument (Dataphysics, Germany). Field emission scanning electron microscope (FE-SEM) observation was conducted on SU8220 (Hitachi, Japan). Biological samples were fixed with 4% paraformaldehyde and dehydrated before observation. The pore size analysis was determined using ASAP2020 (ASAP2020, Micromeritics, USA). The zeta potential of the phase transition system was determined using Litesizer 500 (Anton Paar, Austria). The reflection spectrometer (400–900 nm) (Ideaopitc instruments, NOVA, Shanghai) was employed to detect the thickness of the coating on the silicon wafer[68]. A tensile testing machine (IBTC-300S, Tianjin Care Measure & Control Co., Ltd, China) was used for the bending test and uniaxial tensile test.

## Statistical analysis

All data are mean ± S.D. Student's $t$-test (two group comparison), one-way ANOVA, or two-way ANOVA (multiple groups comparison) were performed to determine statistical significance between different groups. Statistical tests and exact $P$-values used for each data are presented in the figure and legend. Statistics were calculated using GraphPad Prism 9.

## Reporting summary

Further information on research design is available in the Nature Portfolio Reporting Summary linked to this article.

## Data availability

The data supporting the findings from this study are available within the Article, Supplementary Information, or Source Data file. Source data are provided with this paper.

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

## Acknowledgements

We are grateful for funding from the National Science Fund for Distinguished Young Scholars (No. 52225301 to P.Y.), the National Key R&D Program of China (nos. 2020YFA0710400 and 2020YFA0710402 to P.Y.), the National Natural Science Foundation of China (nos. 82070716 to T.C.), the 111 Project (no. B14041 to P.Y.), the Innovation Capability Support Program of Shaanxi (no. 2020TD024 to P.Y.), Science and

Technology Innovation Team of Shaanxi Province (2022TD-35 to P.Y.), and the International Science and Technology Cooperation Program of Shaanxi Province (No. 2022KWZ-24 to P.Y.). We thank the National Local Joint Engineering Research Center for Precision Surgery and Regenerative Medicine (Department of Hepatobiliary Surgery, First Affiliated Hospital of Xi'an Jiaotong University) for their assistance with the experiment.

## Author contributions

T.C. and P.Y. conceived the original concept and initiated this project. J.T. designed the experiment, participated in the entire project, and wrote the article. D.F., Y.X., Y.G., and S.H. assisted in the establishment of primary cell lines and animal experiments. S.M. and K.C. characterized the mechanical properties of the nanofilms. T.C., P.Y., Y.L., Y.Z., and L.X. provided critical feedback and helped revise this article.

## Competing interests

The authors declare no competing interests.
