## [Peer Review File · Nature Communications]

Rectifying disorder of extracellular matrix to suppress urethral stricture by protein nanofilm-controlled drug delivery from urinary catheterReviewers' Comments:

Reviewer #1:

Remarks to the Author:

In the clinic, delivery of numerous antifibrotic drugs by irrigation or submucosal injection showed the low effectiveness against urethral stricture. Here, the authors raised a strategy for both antibacterial therapy and maintenance of extracellular homeostasis for urethral injury. In general, this work seems to be quite interesting and I would like to recommend the acceptance of this work after authors can well address the following questions before its final acceptance.

- 1) Author mentioned that the HE staining in the control group showed abundant inflammatory cell infiltration of the urethra (Fig. 3j). Since HE images could not differentiate the type of cells, the quantification of inflammatory cells (Fig. 3k) should be performed with specific antibodies using ELISA kit.
- 2) Author explained that at the end of catheterization period, the surface of pristine SR catheter was contaminated by the dense *E. coli* biofilm (Figure 3h). However, there is no qualification and quantification of biofilm formation assay (e.g. crystal violet) in this study.
- 3) Author initially seeded the bacteria on the PTB@SA Rapamycin catheters and performed in vitro culture for 20 hours before implantation. However, during in vitro experiment, several parts of bacteria will be eliminated. Since the number of bacteria implanted was different, it will affect the in vivo result, such as inflammatory response, etc. How do the authors explain this finding? Also, the CFU of bacteria after in vivo implantation should be quantified to check its antibacterial efficacy in vivo.
- 4) The rabbit implanted with PTB@SA Rapamycin catheters was not swelling (edema) compared with control. Therefore, the quantification scoring system to check the significance of edema should be performed to support this sentence.
- 5) According to the Figure 6d and s36, there was an obvious change in epithelium thickness between different group, especially in the burst-releasing group. Please explain the correlation of each group with the epithelium thickness.
- 6) According to the Figure 6d, the nuclei in α -SMA staining was obviously observed only in group i, while the other group showed no or very few of nuclei. Meanwhile, in MMP-1 and collagen I staining, nuclei were highly observed in all of the groups. Please explain.
- 7) Antibiotic resistance often occurs in clinical application. Since antibiotic is also used in this system, please explain whether this material could avoid the antibiotic resistance after long-term implantation in vivo.
- 8) Author should add the explanation about the impact of the presence of *E. coli* on urethral fibroblast in co-culture experiment.
- 9) The images of sections in Fig. S37 are not clear. Clear images at a high magnification should be provided.
- 10) Author mentioned that the body weight of the rabbits in each group showed a consistent slight drop within two weeks after urethral coagulation, and then gradually recovered, which may be related to surgical stress (supplementary figure 41). However, there is no statistical analysis that supports the significant difference in body weight.

Reviewer #2:

Remarks to the Author:

This study was designed to investigate the efficacy of rapamycin sustained release nanofilm coated catheters in preventing urethral stricture in a rabbit urethral injury model.

In addition, the researchers have conducted basic experiments on the antibacterial and drug-release effects of catheters, which we believe are appropriate.

However, there are several points that should be considered to improve this study in animal experiments.

1, In the introduction, the author states that local therapy in the urethra has limited efficacy and feasibility in terms of drug retention and effectiveness. However, paclitaxel-coated balloon dilation therapy has already been reported in practice in human patients with anterior urethral strictures and has been shown to be useful and safe (The Journal of Urology, Vol. 207, 866-875, April 2022). The authors should mention this.

2, This study investigates the effect of stricture prevention after injury to the normal urethra. In other words, the results of this study show only a stenosis-preventive effect on a model of 'urethral injury' not 'urethral stricture'. Urethral stricture after urethral injury and restenosis after treatment of urethral stricture have completely different etiologies. The cited paper confuses the paper on the effect of prevention of stricture after urethral injury with the paper on the effect of prevention of restenosis after treatment of urethral stricture, and the authors should consider each separately.

3, Many methods have been reported for creating a urethral stricture/injury model. The author has shown stenosis in a control group by impairing it with electrocautery, is this a well-established method? Also, the author divides the rabbits into 6 groups of 3 rabbits each to evaluate the effect of the drug-releasing catheter. Is three rabbits a sufficient number of rabbits to evaluate this experiment?

4, The author performed urethrography and histological evaluations of the rabbit urethra 30 days later. Were these evaluations performed immediately after removal of the urethral catheter? Immediately after removal of a urethral catheter is not an appropriate time for evaluation of stricture because of the residual effects of physical dilatation caused by catheter placement. In humans, it has been reported that it takes several months for fibrosis of the urethra to be completed (Urology, Vol. 77, 1477-1481, June 2011). Even in animals, it should be evaluated at least several weeks after catheter removal.

Reviewer #3:

Remarks to the Author:

Through this manuscript, the research team reported a novel nanofilm that is used to coat urinary catheter. This nanofilm self-assembled through the deposition of aggregated BSA together with sodium alginate (SA). The aggregation was triggered by a process, termed phase transition bovine serum albumin (PTB). This nanofilm sustained a constant (zero-order) release of encapsulated drug, rapamycin, with in 50 days of observation. The nanofilm delivered rapamycin significantly inhibited fibrosis and urethral stricture as compared to non-coated urinary catheter, systemically administered rapamycin, and other control treatment. This work has immediate clinical significance and profound technical novelty. The characterizations of nanofilm, drug release, and related efficacy are thorough and sound especially the study design for drug release. The description of methods are clear and have sufficient details to reproduce results. Overall, this manuscript merit the acceptance for publication on this journal after minor revision and clarification for the following points.

1, A revision of abstract might be beneficial as the current version does not highlight the important

results and conclusions sufficiently.

2, The choice of BSA as the main materials of the nanofilm may need additional justification. The immunogenicity of BSA to human needs to be considered and discussed. If there is evidence the BSA in nanofilm will not induce humoral responses in human and other species, please provide evidence and reference to reinforce this point.

3. It is unclear whether PTB would only work for BSA, or any proteins as long as they can complete an alpha-helix to beta-sheet conversion. Particularly, it is intriguing whether the PTB will work with human albumin?

4, The CD data that support helix-to-sheet conversion is not sufficient. It is not convincing the end product is a beta-sheet-only material with only CD spectra. Though, whether this claim is true or not seems not affecting functionality of the nanofilm.

5, On page 14, it is not clear why a balanced composition of positively and negatively charged amino acid residues is critical for anti-fouling and anti-biofilm surface.

6, On page 25, this description is confusing "Compared with the unmodified catheter, the anti-fibrotic catheter (i.e., PTB@SA-Rapamycin catheter) decreased lumen reduction from 56.1% to 9.8% (Fig. 6a, b), with a mean percentage improvement of 82.5%, which was superior to the values reported in preclinical studies of other local treatments (Fig. 6c, Table S5)." It is suggested to break this description into at least two sentences.

7, For efficacy study (urethral stricture), it would be beneficial compare lumen diameters of subgroup VI with healthy (non-injuries) rabbit lumen diameters to give a complete picture of treatment outcome. In addition, the group size (n=3) is too small to generate robust mean and conclusion.

Reviewer #4:

Remarks to the Author:

Comments to the author

The authors of this manuscript have done an excellent work designing a catheter covered with a protein nanofilm capable of preventing urethral stricture by prevention of bacterial colonization and controlled release of rapamycin. The authors examined the physical characteristics, antibacterial properties, and drug release functions of the protein nanofilm. By indwelling the hybrid coating-modified catheter in the urethra of adult male rabbits, the authors demonstrated that the catheter could effectively prevent the development of urethral stricture.

Generally, this work is innovative and great. A few comments are as follows:

Q1

In the first paragraph of the introduction, the author pointed out a clinical problem that current curative treatment options, such as urethrotomy, dilatation and urethroplasty, for urethral stricture could not achieve satisfactory results. This clinical problem exists in patients who have already developed severe urethral strictures. However, the animal model used in this study only simulated the acute urethral injury, and the results could only demonstrate the fact that PTB@SA-Rapamycin catheter was able to prevent the occurrence of urethral stricture. Therefore, the clinical problem raised in the introduction should be revised.

Q2

Why did the authors choose rapamycin as the therapeutic component preventing the urethral stricture? A meta-analysis published in European Urology by Pang et al. showed that mitomycin C had

the best effect on urethral stricture among drugs used in the included clinical trials. Compared with other drugs, what are the advantages and disadvantages of rapamycin in treating urethral stricture?

Q3

The author mentioned that the nanofilm surface exhibited abundant functional groups, which enabled the nanofilms to undergo ligand bonding, electrostatic interaction, hydrophobic interaction and hydrogen bonding with virtually arbitrary material, including metal, organic or inorganic materials. According to our experience, this property of the nanofilm makes it likely to interact with various organic and inorganic substances in urine and result in surfaces blockade, stone formation and catheter blockage. Is there any data showing the appearance of the hybrid nanofilm-coated catheters after indwelling for 30 days? We want to check if the surface of all catheters were still clear at the end of the experiment. In fact, bacterial contamination and stone formation are both thorny challenges in the development of functional urinary catheters.

Q4

The nanofilm showed a consistent WCA approximately 80°. This means the nanofilm-coated catheter is not very hydrophilic and has higher friction with urethral. Was lubricant used during indwelling catheters. If lubricant was used, were there any experiments investigating if the lubricant had any effect on the stability of the nanofilm and the controlled drug release? Catheters for clinical use are often covered with super hydrophilic film or lubricants to avoid urethral injury and inflammation. Consistently, the authors observed that the pristine urinary catheter caused edema and necrosis in rabbits' urethral mucosa after indwelling for 1 month. Parameters about the friction should be compared among pristine urinary catheter and nanofilm coated catheters.

Q5

Reference should be added for the sentence "... the PTB@SA nanofilm still exhibited a low negative surface zeta potential, and thus may keep a good anti-biofilm ability."

Q6

We noticed that four rabbits were in each group for the evaluation of the antibacterial biofilm performance of the nanofilm and three rabbits were in each group evaluate the effect of nanofilm on urethral stricture. How did the authors decide the sample size for each experiment? Why did experiments with more complex surgical procedures (more likely to be biased) have smaller sample sizes?

Q7

The authors cannot directly compare the results of independent experiments performed by different operators under different experimental conditions (such as surgical instruments, degree of urethral injury and therapy duration) to conclude that the effect of nanofilm in this study was better than that of drugs reported in other studies.

Q8

In the figure legend of figure 6, the sentence "(b, d, e) i, Control, ii; Unmodified catheter; iii,..." should be revised. Only a and d are numbered. Please check the use of commas and semicolons.

Q9

At least, there are some errors in references 56. Some authors are missing and the original title should be "Sirolimus-eluting Biodegradable Poly-l-Lactic Acid Stent to Suppress Granulation Tissue Formation in the Rat Urethra."

Dear reviewers,

We sincerely thank all referees for their valuable comments. We have made the revisions to all the comments in the revised manuscript (marked in red color) and supporting information (SI). The following is the point-to-point response to the reviewers' comments.

Reviewer #1

In the clinic, delivery of numerous antifibrotic drugs by irrigation or submucosal injection showed the low effectiveness against urethral stricture. Here, the authors raised a strategy for both antibacterial therapy and maintenance of extracellular homeostasis for urethral injury. In general, this work seems to be quite interesting and I would like to recommend the acceptance of this work after authors can well address the following questions before its final acceptance.

1) Author mentioned that the HE staining in the control group showed abundant inflammatory cell infiltration of the urethra (Fig. 3j). Since HE images could not differentiate the type of cells, the quantification of inflammatory cells (Fig. 3k) should be performed with specific antibodies using ELISA kit.

Response: We are extremely grateful to Reviewer 1 for your helpful comments. Through H&E staining, we found that compared with the control group, the infiltrated area of cells with high nucleocytoplasmic ratio cells in the urethral tissue of the PTB@SA coating group was significantly reduced, suggesting that the distribution of inflammatory cells between the two groups may be different. To better define and localize the infiltration of inflammatory cells in urethral tissue, we performed immunochemical staining with CD45 (expressed in leukocytes) and CD68 (expressed in macrophages) antibodies. As shown in Fig. 3k, in addition to a significant decrease in the area of positive cells, inflammatory cells (including both leukocytes and macrophages) were confined to the epithelial mucosa of urethral tissues in the PTB@SA nanofilm-modified catheter group, while a large number of inflammatory cells were diffusively infiltrated in the mucosa and submucosa of the unmodified catheter group. The above discussion has been incorporated into the revised manuscript (Page 17).

Figure. 3 The anti-biofilm performance and biocompatibility of the hybrid coating.

a, Zeta potential of the PTB or PTB@SA(0.10) coating on Si. Data are mean \pm s.d.; n=3. **b, c**, Number of viable *S. aureus* (**b**) and *E. coli* (**c**) recovered from bare substrate, PTB or PTB@SA(0.10) coating as confirmed by colony counting. Data are mean \pm s.d.; n=3. **d, e**, CLSM images showing the patterned adherence of *S. aureus* on bare substrate with using the micropatterned PTB@SA coating as the resistant layer. **f, g** SEM images of unmodified (**f**) or PTB@SA(0.10) coating-modified urinary catheters (**g**) recovered after indwelling for 1 week. The arrow indicates scattered bacteria on the PTB@SA(0.10) coating. **h**, In vivo anti-biofilm efficiency of the hybrid coating measured by colony counting. Data are mean \pm s.d.; n=4. **i**, The number of white blood cells in rabbit urine after different catheters were used for 1 week. Data are mean \pm s.d.; n=4. **j**, Gross observation of the rabbits' urethra, and the yellow circles represent the sampling site of the tissue section. **k**, Hematoxylin and eosin (H&E) staining and immunochemical staining with anti-CD45 antibody (green) and anti-CD68 (brown) antibody in rabbit urethral tissue. **l**, Quantification of the

area of inflammatory cell infiltration in the H&E staining images of the urethra. Data are mean \pm s.d.; n=4. **m**, Quantification of CD45 (l) and CD68 (m) positive area in **Fig. 3k**. Data are mean \pm s.d.; n=4. * P<0.05, ** P<0.01, *** P<0.001, ns means no statistical difference.

2) Author explained that at the end of catheterization period, the surface of pristine SR catheter was contaminated by the dense *E. coli* biofilm (Figure 3h). However, there is no qualification and quantification of biofilm formation assay (e.g. crystal violet) in this study.

Response: Thank the reviewer for this comment. Since crystal violet also stains the protein coating to bias the optical density of the sample¹, we alternatively used the number of bacteria per unit catheter surface area to evaluate the *in vivo* anti-biofilm performance of the PTB@SA coating. After the catheter was indwelled in the rabbit urethra for 7 days, the surface bacterial density was determined by colony counting. Compared with the control group, the resistance efficiency of the PTB@SA coating against *E. coli* was 85.9%, which is consistent with the SEM image results of the recovered catheter (Fig. 3h in the revised manuscript).

Figure. 3 The anti-biofilm performance and biocompatibility of the hybrid coating

h, *In vivo* anti-biofilm efficiency of the hybrid coating measured by colony counting. Data are mean \pm s.d.; n=4. ** P<0.01.

3) Author initially seeded the bacteria on the PTB@SA Rapamycin catheters and performed *in vitro* culture for 20 hours before implantation. However, during *in vitro* experiment, several parts of bacteria will be eliminated. Since the number of bacteria implanted was different, it will affect the *in vivo* result, such as inflammatory response, etc. How do the authors explain this finding? Also, the CFU of bacteria after *in vivo* implantation should be quantified to check its antibacterial efficacy *in vivo*.

Response: Thank the reviewer for this comment. Here, we adopted the pre-seeding method, a typical method widely used to characterize antifouling property in animal experiments^{2,3}, to evaluate the efficacy of the PTB@SA coating in preventing catheter-associated urinary tract infections. In fact, microorganisms transferred from the perianal or skin during catheterization, and/or pathogenic bacteria migrating along the luminal pathway have been reported to be important sources of infection for catheter-associated urinary tract infections⁴. Bacterial biofilms developed from bacteria that colonize on the material surfaces not only provide shelter for bacteria but also participate in

quorum-sensing signaling⁵. As a result, compared with planktonic bacteria, bacteria surviving in biofilms exhibit strong resistance to antibiotics and mechanical challenges, which will cause refractory and recurrent urinary tract infections. Therefore, in order to reasonably simulate the bacterial contamination that catheters may be exposed to in practical applications and evaluate the consequences caused in vivo, unmodified urinary catheters and PTB@SA nanofilm-modified catheters were all incubated in artificial urine containing *E. coli* at a consistent final bacterial concentration of 10^6 CFU/mL, and all other experimental conditions were the same for both groups. The surface properties of the catheter modified by the PTB@SA coating make it difficult for bacteria to attach stably, so the coating exhibits an ideal preventive effect on catheter-associated urinary tract infection. The bacterial load of the recovered catheters of the two groups was evaluated by colony counting. As shown in Fig 3h, the number of bacteria on the surface of the PTB@SA coating was significantly lower than that of the control group ($P < 0.01$).

Figure. 3 The anti-biofilm performance and biocompatibility of the hybrid coating

h, In vivo anti-biofilm efficiency of the hybrid coating measured by colony counting. Data are mean \pm s.d.; n=4. ** $P < 0.01$.

4) The rabbit implanted with PTB@SA Rapamycin catheters was not swelling (edema) compared with control. Therefore, the quantification scoring system to check the significance of edema should be performed to support this sentence.

Response: Thanks to the reviewer for your helpful comments. In response to this comment, edema was scored semiquantitatively on a 3-point scale in tissue sections, with 0 being the lowest and 2 being the highest scores, as previously described⁶. As shown in Supplementary Fig. 21, consistent with the macroscopic appearance of the specimens, the urethral tissue of the control group had a higher edema score compared with the PTB@SA-coated catheter ($P < 0.01$). The above discussion has been incorporated into the revised manuscript (Page 17).

Figure S21. Quantitative assessment of the edema severity in control and PTB@SA-coated groups. Data are presented as the mean \pm s.d.; n=4. ** P<0.01.

5) According to the Figure 6d and s36, there was an obvious change in epithelium thickness between different group, especially in the burst-releasing group. Please explain the correlation of each group with the epithelium thickness.

Response: Thanks for the reviewer's comment. We carefully compared the overall condition of the rabbit urethral tissue sections in each group and concluded that the thickness of the urothelium in each group was not related to the treatment received, but the thickness and collagen abundance of submucosal connective tissue in the presence of rapamycin sustained-release catheter were significantly lower than other groups, as shown in Figure 6 and S38. In fact, on the one hand, the histological morphology of the urethral epithelium is related to the location of the specimen. The urethral injury site in this experiment was about 1 cm distal to the opening of ejaculator duct, which was exactly at the position where the prostatic urethra (mainly transitional epithelium) transitioned to the membranous urethra (typically stratified columnar epithelium)⁷. On the other hand, even for different lateral walls of the urethra in the same cross-section, the thickness of the epithelium varies slightly, as has been observed in the published literature (Figure 3, 4 of the reference)⁸.

6) According to the Figure 6d, the nuclei in α -SMA staining was obviously observed only in group i, while the other group showed no or very few of nuclei. Meanwhile, in MMP-1 and collagen I staining, nuclei were highly observed in all of the groups. Please explain.

Response: We greatly appreciate this helpful comment. In the α -SMA-stained tissue, the nuclei were indeed not clearly stained because of a mistake in our staining procedure. For this, we re-stained and analyzed accordingly. The corresponding results have been revised in Fig 6d and 6e.

Figure. 6 The sustained-release coating of rapamycin on the urinary catheter attenuates collagen deposition and inhibits the formation of strictures post urethral injury.

a, Representative retrograde urethrogram of rabbits with different treatments after urethral electrocoagulation, using 76% Meglumine Diatrizoate as a contrast agent. **b**, Quantitative analysis of urethral lumen reduction and submucosal collagen density in rabbits with different treatments.

Data are mean \pm s.d.; n=5. **c**, Corresponding results of the PTB@SA-Rapamycin catheter compared with other preclinical studies of topical therapy to suppress urethral stricture. **d**, Gross specimen, H&E staining and immunohistochemical staining with anti-alpha smooth muscle actin antibody (α -SMA, brown), anti-matrix metalloproteinase 1 antibody (MMP1, brown), and type I collagen antibody (Collagen I, green) of the urethral injury sites at 1 month. Nuclei (blue) were stained with DAPI. Yellow triangles indicate urethral scar tissue. The asterisks represent urethral luminal side, and the dotted lines indicate the boundary between the submucosa and the muscularis. **e**, Quantification of α -SMA, MMP1 positive area and Collagen I fluorescence intensity in **Fig. 6d**. Data are mean \pm s.d.; n=5. **f, g**, Hematological examination to show the changes of alanine aminotransferase (ALT, **f**) and aspartate aminotransferase (AST, **g**) with different treatments for 1 month. Data are mean \pm s.d.; n=5. **h**, H&E staining of rabbit liver after systemic administration or indwelling rapamycin sustained-release catheter for 1 month. (**a, d**) **i**, Control; **ii**, Unmodified catheter; **iii**, Systemic administration; **iv**, PTB@SA; **v**, Burst-releasing; **vi**, PTB@SA-Rapamycin. * P<0.05, ** P<0.01, *** P<0.001, **** P<0.0001, ns means no statistical difference.

7) Antibiotic resistance often occurs in clinical application. Since antibiotic is also used in this system, please explain whether this material could avoid the antibiotic resistance after long-term implantation in vivo.

Response: Thanks for the reviewer's comment. Here, we need to clarify that the antibacterial performance of the functionalized urinary catheter is mainly derived from the excellent antibacterial adhesion property of the PTB@SA nanofilm, and the rapamycin contained in the system is not a typical antibiotic, so it will not induce bacteria drug resistance. As a classic mTOR pathway inhibitor, rapamycin is widely used in the prevention of coronary artery stenosis in clinical practice, and shows favorable biological safety⁹. Inspired by this, in the present work, controlled release of rapamycin from the coating on the catheter surface was the main contributor to the inhibition of stricture formation after urethral injury, and the effect of preventing catheter-related urinary tract infections depended on the anti-biofilm property of the hybrid coating. The above discussion has been incorporated into the revised manuscript (Page 15).

8) Author should add the explanation about the impact of the presence of E. coli on urethral fibroblast in co-culture experiment.

Response: Thanks for the reviewer's comment. The proliferation and activation of fibroblasts is a key link in the occurrence and development of urethral stricture¹⁰. The inflammatory response caused by bacteria induces changes in the biological behavior of fibroblasts through a variety of cytokines (such as TGF- β 1, TNF- α , IL-6, etc.)¹¹, which eventually promotes the formation of urethral strictures and even leads to the failure of urethroplasty¹². As shown by CD45 and CD68 staining of urethral tissue, unmodified catheters were more susceptible to biofilm contamination and urethral inflammation than the coated group (Fig. 3k-m). Therefore, the PTB@SA nanofilm with potent anti-biofilm property is a strong candidate for designing drug-loaded coatings on the surface of urinary catheters to regulate tissue healing, which can effectively attenuate the fibrotic process initiated by inflammation. The above discussion has been incorporated into the revised manuscript (Page 17).

9) The images of sections in Fig. S37 are not clear. Clear images at a high magnification should

be provided.

Response: Thanks for the reviewer's comment. We have supplied clear images at high magnification to the revised supplementary information (Page S30, Supplementary Fig. 41).

Figure S41. Immunofluorescence staining images showing collagen deposition in urethral tissue. (a) Immunofluorescence staining of Collagen I at the urethral injury site after different treatments for 1 month. Nuclei (blue) was stained with DAPI. i, Control; ii, Unmodified catheter; iii, Systemic administration; iv, PTB@SA; v, Burst-releasing; vi, PTB@SA-Rapamycin. (b) The urethral tissue of normal rabbits was sectioned for immunostaining with Collagen I. Nuclei (blue) was stained with DAPI. (c) Quantification of the Collagen I fluorescence intensity in normal rabbit urethral tissue and PTB@SA-Rapamycin catheter-mediated urethral injury healing. Data are

presented as the mean \pm s.d.; n=5. ns means no statistical difference.

10) Author mentioned that the body weight of the rabbits in each group showed a consistent slight drop within two weeks after urethral coagulation, and then gradually recovered, which may be related to surgical stress (supplementary figure 41). However, there is no statistical analysis that supports the significant difference in body weight.

Response: Thanks to the reviewer for this valuable comment. To support our conclusion, one-way ANOVA was performed on the body weight of each group at each time point to confirm that there was no statistical difference in the body weight of rabbits in each treatment group and the blank control group. We have supplied a related description in Supplementary Fig. 45.

Figure S45. The initial body weight and the weight of the rabbits in each group at 1, 2, 3 and 4 weeks after the urethra was injured by electrocoagulation. The data are presented as the mean \pm s.d.; n=5. At each time point, there was no statistical difference in body weight between the treatment groups and the blank control group.

Reviewer #2

This study was designed to investigate the efficacy of rapamycin sustained release nanofilm coated catheters in preventing urethral stricture in a rabbit urethral injury model. In addition, the researchers have conducted basic experiments on the antibacterial and drug-release effects of catheters, which we believe are appropriate. However, there are several points that should be considered to improve this study in animal experiments.

1, In the introduction, the author states that local therapy in the urethra has limited efficacy and feasibility in terms of drug retention and effectiveness. However, paclitaxel-coated balloon dilation therapy has already been reported in practice in human patients with anterior urethral strictures and has been shown to be useful and safe (The Journal of Urology, Vol. 207, 866-875, April 2022). The authors should mention this.

Response: Thanks to the reviewer for your helpful comments. For patients with urethral strictures, paclitaxel-coated balloons are indeed a beneficial option that combines mechanical dilation and local drug delivery, indicating that local therapy to intervene in the progression of urethral strictures is a feasible and promising strategy. Here, we attempted to combine intraurethral drug delivery with widely used urinary catheters to develop functionalized urinary catheters with the anti-fibrotic effect, which could hopefully evolve treatment after urethral strictures to positive prophylaxis, especially for strictures caused by transurethral procedures and traumatic catheterization. We have supplied relevant content in the Introduction section (Page 3) as follows.

Although systemically administered antifibrotic agents have shown efficacy in attenuating tissue/organ fibrosis in animal models^{13,14}, off-target side effects have greatly limited their success in clinical trials¹⁵. By contrast, the successful attempt of a paclitaxel-coated balloon combining mechanical dilation and local drug delivery in the treatment of recurrent urethral strictures suggests the feasibility and promise of local therapy to interfere with urethral strictures¹⁶. Currently, in order to inhibit secondary strictures after urethral injury, various antifibrotic drugs have been administered locally by hydrostatic pressure, submucosal injection, urethral irrigation, drug-eluting stents and catheters¹⁷⁻²¹.

2, This study investigates the effect of stricture prevention after injury to the normal urethra. In other words, the results of this study show only a stenosis-preventive effect on a model of ‘urethral injury’ not ‘urethral stricture’. Urethral stricture after urethral injury and restenosis after treatment of urethral stricture have completely different etiologies. The cited paper confuses the paper on the effect of prevention of stricture after urethral injury with the paper on the effect of prevention of restenosis after treatment of urethral stricture, and the authors should consider each separately.

Response: Thanks to the reviewer for your professional comments. Admittedly, the results of animal experiments provide strong support for the conclusion that the anti-fibrotic catheter has a good preventive effect on urethral stricture secondary to urethral injury. We apologize for the ambiguity caused by our inappropriate description and have revised the Introduction as follows to emphasize that this study addresses urethral strictures secondary to urethral injury (Page 2).

Urethral stricture is a common disease (200–1200 cases per 100000 individuals²²) following urethral injury. The pathological state of extracellular matrix (ECM) metabolic disorder caused by injury (including surgery, pelvic fracture, inflammatory injury and traumatic catheterization) results

in the replacement of normal urethral tissue by dense fibers interspersed with fibroblasts²². This aggravated fibrosis eventually leads to progressive urethral lumen reduction consequent symptomatic dysuria and even renal impairment. With the growing demand for healthcare brought about by the aging population and advances in medical technology, the increasing incidence of mucosal injury and secondary urethral strictures caused by various transurethral procedures has attracted great attention^{23,24}. Especially in patients with spinal cord injury or in intensive care, urethral stricture or erosion due to long-term catheterization has been reported as high as 8.7%²⁵. Once urethral injury progresses to urethral stricture, subsequent treatment is extremely troublesome, which will pose challenges to both patients and urologists. The long-term success rate of the most commonly used endoscopic treatment is only 20~30%²², and although urethroplasty has developed rapidly in recent years, fibrous scarring may still develop in the urethral submucosa after substitute surgery, and this open surgery is less suitable for the elderly and frail patients. Indeed, the pathological state of ECM metabolic disorder and subsequent scar repair is not be rectified regardless of the treatment²⁶. Therefore, there is an urgent need to develop alternative strategies other than surgery and attempt to focus on regulating healing by inhibiting fibrosis progression following urethral injury, thereby preventing urethral strictures.

3, Many methods have been reported for creating a urethral stricture/injury model. The author has shown stenosis in a control group by impairing it with electrocautery, is this a well-established method? Also, the author divides the rabbits into 6 groups of 3 rabbits each to evaluate the effect of the drug-releasing catheter. Is three rabbits a sufficient number of rabbits to evaluate this experiment?

Response: Thanks to the reviewer for your helpful comments. There are a variety of modalities commonly used to cause urethral injury in animals, mainly including urethrotomy, electrocoagulation, and ligation. According to a study reported by the *Journal of Urology* in 2012, among the three injury methods mentioned above, electrocoagulation showed more enhanced fibrosis with the highest similarity to the human stricture specimens²⁷. In addition, our group has rich experience in the model of urethral injury caused by electrocoagulation, and the outcome and pathological changes of urethral stricture caused by this method were studied in detail²⁸. The results showed that all rabbits subjected to electrocoagulation developed significant urethral stricture, and the tissue of the stricture site showed typical fibrosis characteristics. In view of the above reasons, electrocoagulation was selected as an effective method for the formation of urethral injury in this study, and the effect of different treatments was further evaluated.

Regarding the sample size of each group, we performed animal experiments to evaluate the effect of the PTB@SA-Rapamycin catheter in a larger sample size (n=5) and revised the updated results in the revised manuscript. Representative images of retrograde urethrography are shown below (Fig. R1).

FigureR1. Representative images of retrograde urethrography after 30 days. The yellow arrow indicates the coagulation site.

4, The author performed urethrography and histological evaluations of the rabbit urethra 30 days later. Were these evaluations performed immediately after removal of the urethral catheter? Immediately after removal of a urethral catheter is not an appropriate time for evaluation of stricture because of the residual effects of physical dilatation caused by catheter placement. In humans, it has been reported that it takes several months for fibrosis of the urethra to be completed (Urology, Vol. 77, 1477-1481, June 2011). Even in animals, it should be evaluated at least several weeks after catheter removal.

Response: Thanks to the reviewer for this professional comment. To respond this comment, we then supplemented the animal experiment to perform retrograde urethrography and histological evaluation at 2 weeks after catheter removal (Supplementary Fig. 40). Being similar to the evaluation conclusion from the case of immediate group, at 2 weeks after catheter removal, the degree of urethral lumen reduction in the PTB@SA-Rapamycin group remained significantly lower than that of the unmodified catheter (Supplementary Fig. 40 a, b). The urethral gross specimens in the unmodified catheter group showed white urethral scar and abundant collagen fibers in the submucosa, while the urethral mucosa in the PTB@SA-Rapamycin group was smooth and ruddy, and correspondingly the density of collagen in the submucosa was also reduced (Supplementary Fig. 40 c, d). Furthermore, α -SMA, a marker of myofibroblast activation, is important in predicting the clinical progression of fibrotic diseases including liver fibrosis and urethral strictures^{29,30}. After 30 days of catheterization and 2 weeks after catheter removal, the expression of α -SMA, in the healed urethral tissue treated with anti-fibrotic catheter was significantly and consistently lower than that in the unmodified catheter group, indicating that fibroblast activation and subsequent ECM synthesis were inhibited in the PTB@SA-Rapamycin group, which also suggested that the risk of disease progression might be reduced.

Overall, our experimental results suggest that the anti-fibrotic catheter can effectively regulate ECM metabolism in the process of injury healing to reduce the deposition of excess ECM in the submucosa, and ultimately successfully intervene in the formation of stricture after urethral injury. This functional urinary catheter may provide a prophylactic and beneficial option for those at high risk of urethral strictures, especially those undergoing traumatic catheterization and transurethral procedures.

Figure S40. Rabbit urethral stricture in unmodified urinary catheter group and PTB@SA-Rapamycin group on the 14th day after removal of catheter. (a) Representative retrograde urethrogram using 76% meglumine diatrizoate as contrast agent. (b) Lumen reduction immediately after catheter removal or 2 weeks later as determined by retrograde urethrography. Data are mean \pm s.d.; n=5. (c) Comparison of gross specimens, H&E staining, Masson staining and α -SMA expression between unmodified urinary catheter group and PTB@SA-Rapamycin group at 14 days after catheter removal. Yellow triangles indicate urethral scar tissue, and the dotted lines indicate the boundary between the submucosa and the muscularis. (d, e) Quantitative results of collagen density and α -SMA positive area in Figure S40c. Data are mean \pm s.d.; n=5. *** P<0.001.

Reviewer #3

Through this manuscript, the research team reported a novel nanofilm that is used to coat urinary catheter. This nanofilm self-assembled through the deposition of aggregated BSA together with sodium alginate (SA). The aggregation was triggered by a process, termed phase transition bovine serum albumin (PTB). This nanofilm sustained a constant (zero-order) release of encapsulated drug, rapamycin, within 50 days of observation. The nanofilm delivered rapamycin significantly inhibited fibrosis and urethral stricture as compared to non-coated urinary catheter, systemically administered rapamycin, and other control treatment. This work has immediate clinical significance and profound technical novelty. The characterizations of nanofilm, drug release, and related efficacy are thorough and sound especially the study design for drug release. The description of methods are clear and have sufficient details to reproduce results. Overall, this manuscript merit the acceptance for publication on this journal after minor revision and clarification for the following points.

1, A revision of abstract might be beneficial as the current version does not highlight the important results and conclusions sufficiently.

Response: Thanks for this valuable comment. We have revised the Abstract in the manuscript as follows to emphasize the significance of our study.

Urethral stricture secondary to urethral injury, afflicting both patients and urologists, is initiated by excessive deposition of extracellular matrix in the submucosal and periurethral tissues. Although various anti-fibrotic drugs have been applied to urethral stricture by irrigation or submucosal injection, their clinical feasibility and effectiveness are limited. Here, to target the pathological state of the extracellular matrix, we designed a protein-based nanofilm-controlled drug delivery system and assembled it on the catheter. This approach, which integrates excellent anti-biofilm property with stable and controlled drug delivery for tens of days in one step, ensures optimal efficacy and negligible side effects while preventing biofilm-related infections. In a rabbit model of urethral injury, the anti-fibrotic catheter maintains extracellular matrix homeostasis by reducing fibroblast-derived collagen production and enhancing metalloproteinase 1-induced collagen degradation, resulting in the greatest improvement in lumen stenosis than other topical therapies for urethral stricture prevention. Such facily fabricated biocompatible coating with antibacterial contamination and sustained-drug-release functionality could not only benefit populations at high risk of urethral stricture, but also serve as an advanced paradigm for a range of biomedical applications.

2, The choice of BSA as the main materials of the nanofilm may need additional justification. The immunogenicity of BSA to human needs to be considered and discussed. If there is evidence the BSA in nanofilm will not induce humoral responses in human and other species, please provide evidence and reference to reinforce this point.

Response: Thanks to the reviewer for your helpful comments. The readily available BSA was chosen as the main material for the nanofilm because it is a natural polymer derived from the organism and has natural anti-adsorption property. The anti-biofilm and anti-protein adsorption properties of the macroscopic PTB nanofilm transformed from native BSA under the induction of TCEP were further improved (indicated by higher efficiency and greater stability)³¹, which are desirable surface characteristics for various medical devices including urinary catheters. Based on

the above, BSA is widely used in the field of designing new wound dressings and shows no biosafety concerns^{32,33}. For the coating of surface devices such as urinary catheters, as long as the BSA in the coating does not penetrate into the human body, it is unlikely to activate immune cells and further trigger an immune response.

As previously demonstrated by our group, the PTB-based cream neither irritated the rabbit skin nor penetrated the skin to cause potential side effects³⁴. The subcutaneous implantation experiment also showed that the PTB coating did not induce foreign body giant cell reaction or inflammatory necrosis³¹, which is consistent with the results of the intraurethral implantation experiment reported in this work (Supplementary Fig. 19). In addition, human serum albumin (HSA), which is highly similar to BSA in terms of gene sequence, can also react with TCEP to form a macroscopic coating, and the corresponding experimental results are shown in Supplementary Fig. 47. In the future, for implant surface coatings that require long-term direct contact with blood, alternatively using HSA instead of BSA to react with TCEP can better meet the higher immunogenicity requirements.

3. It is unclear whether PTB would only work for BSA, or any proteins as long as they can complete an alpha-helix to beta-sheet conversion. Particularly, it is intriguing whether the PTB will work with human albumin?

Response: Thanks to the reviewer for your helpful comments. In fact, similar reactions to form nanofilms by protein self-assembly triggered by reducing agents have been observed in a series of commonly used proteins such as insulin, lysozyme and α -lactalbumin^{35,36}. In our previous study, these proteins that can undergo amyloid-like assembly were identified as proteins with the following 3 characteristics, including (1) a high fibrillation propensity segment, (2) abundant alpha-helices and (3) the reduction of S-S bonds by TCEP³⁶. As the homologous protein of BSA, HSA can also undergo a similar reaction, and the corresponding results are shown in Supplementary Fig. 47. As indicated by AFM, the PTH nanofilm was formed by close packing of nanoscale particles (Supplementary Fig. 47 a). Far-UV circular dichroism (CD) spectra confirmed the loss of α -helix (208 and 222 nm), accompanied by a significant increase in β -sheet (216 nm) in the PTH nanofilm (Supplementary Fig. 47 b). The ThT fluorescence results of the phase transition system reflected the continuous accumulation of β -sheet structure in the reaction system, and the successful staining of the nanofilm with ThT and Congo red also indicated that the resultant nanofilm was rich in β -sheet (Supplementary Fig. 47 c). Based on the above, we further doped sodium alginate (SA) into the phase transition system and confirmed that SA was successfully integrated into the nanofilm (Supplementary Fig. 47 d). Compared with bare Si, both the water contact angle and surface elements changed significantly after incubating with the phase transition solution (Supplementary Fig. 47 e, f), indicating that the PTH or PTH@SA nanofilm was successfully coated on Si. The robust adhesion of the nanofilm originates from various polar or nonpolar functional groups exposed on the nanofilm surface that can interact with the underlying substrate in a variety of ways. As revealed by the high-resolution XPS spectra of C_{1s} of the nanofilm (Supplementary Fig. 47 g), the nanofilm typically presented structures including aliphatic carbon (C-H/C-C), amines (C-N), hydroxyls (C-O), thiols (C-S) amides (O=C-N), and carboxyl groups (O=C-O). To further characterize the adsorption capacity of the samples, solutions with different components were pumped into the chamber of the QCM-D. After 1 h, the adsorption mass of the PTH and PTH@SA nanoparticles on the Au chip exceeded 2000 ng/cm², which was significantly higher than that of HSA and the mixture of HSA and SA (Supplementary Fig. 47 h-k). The above results indicate that

HSA can also undergo a similar phase transition process triggered by TCEP, and the SA-doped polysaccharide-protein composite nanofilm can be easily obtained by adding SA to the phase transition system. The corresponding description and discussion have been incorporated into the revised manuscript (Page 31) and Supplementary Information (Page S35-S36).

Figure S47. Formation and hybridization of PTH the nanofilm. (a) AFM image of the PTH nanofilm. (b) CD spectra of native HSA and the PTH nanofilm. (c) ThT fluorescence change as a function of phase transition time, with the insets showing the corresponding fluorescence microscopic image for ThT staining and the optical microscopy image for Congo red staining. (d) The CLSM image shows that SA was successfully entrapped in the nanofilm to form a hybrid nanofilm. (e) WCA of bare Si, PTH and PTH@SA coated Si. The data are presented as the mean \pm s.d.; n=4. (f) XPS spectra of pristine Si, PTH and PTH@SA nanofilm coated Si. (g) High resolution C_{1s} deconvolution spectra of the PTH and PTH@SA nanofilm. (h-k) The frequency and adsorption mass of HSA (h), PTH (i), the mixture of HSA and SA (j) and PTH@SA (k) adsorbed on the Au chip as a function of time.

4, The CD data that support helix-to-sheet conversion is not sufficient. It is not convincing the end product is a beta-sheet-only material with only CD spectra. Though, whether this claim is true or not seems not affecting functionality of the nanofilm.

Response: Thanks to the reviewer for your helpful comments. We apologize for the ambiguity caused by our inappropriate wording. CD is a spectroscopic technique commonly used to investigate protein secondary structures through characteristic CD spectra produced by major protein secondary structure³⁷. As shown in Fig. 1d, the negative α -helix peak for the native BSA at 208 and 222 nm was shifted to the single negative band at 216 nm for β -sheet structure after the phase transition³⁸, indicating that the TCEP-induced protein unfolding was accompanied by a change in major secondary structure from α -helix to β -sheet, rather than β -sheet being the only protein secondary structure in the nanofilm. In addition, the deconvolution of the amide I and II in the FTIR spectra of the PTB and PTB@SA nanofilm also supported the increased β -sheet structure in nanofilms compared with the native BSA (Supplementary Fig. 2). We have rephrased the corresponding description in the revised manuscript to articulate our point (Page 6).

5, On page 14, it is not clear why a balanced composition of positively and negatively charged amino acid residues is critical for anti-fouling and anti-biofilm surface.

Response: Thanks to the reviewer for your helpful comments. BSA is a macromolecule with natural antifouling properties, which is related to the balance of positive and negative charges on albumin protein surface (the balance of glutamic acid and lysine)^{31,39}. This mixed-charge surface, similar to zwitterionic polymers, can effectively resist contamination by proteins and bacteria through hydration. However, fabricating of BSA coating on multiple classes of materials have been limited by surface chemistry. Based on the mixed-charge principle, some scholars have successfully constructed natural peptide surfaces with antifouling properties by selecting appropriate amino acid residues²⁷. In this study, we used a reducing agent to induce BSA to unfold and further self-assemble into a macroscopic nanofilm, which is a controllable process rather than just non-specific adsorption of proteins on the surface. The resultant biobased coating has a near-neutral surface charge, as reflected by the zeta potential results of the material (Fig. 3a). In this way, such a coating with both hydrogen bonding groups and zwitterionic groups can not only solvate the material through electrostatic interactions, but also form hydrogen bonds with water molecules through peptide bonds⁴⁰, thereby forming a hydration layer on the surface as a barrier to prevent protein or microorganisms from contacting, and finally achieve the effect of anti-protein adsorption and anti-biofilm synergistically. The related discussion has been supplemented and revised in the manuscript (Page 15).

6, On page 25, this description is confusing "Compared with the unmodified catheter, the anti-fibrotic catheter (i.e., PTB@SA-Rapamycin catheter) decreased lumen reduction from 56.1% to 9.8% (Fig. 6a, b), with a mean percentage improvement of 82.5%, which was superior to the values reported in preclinical studies of other local treatments (Fig. 6c, Table S5)." It is suggested to break this description into at least two sentences.

Response: Thanks to the reviewer for the reminder. To make the presentation more concise, we have split this description into the following 2 sentences and amended it based on the results of the supplementary animal experiments. "Compared with the unmodified catheter, the anti-fibrotic

catheter (i.e., PTB@SA-Rapamycin catheter) decreased lumen reduction from 51.8% to 10.6% (Fig. 6a, b). Such a mean percent improvement of up to 79.2% is superior to values reported in preclinical studies of other topical treatments (Fig. 6c, Table S5).”

7, For efficacy study (urethral stricture), it would be beneficial compare lumen diameters of subgroup VI with healthy (non-injuries) rabbit lumen diameters to give a complete picture of treatment outcome. In addition, the group size (n=3) is too small to generate robust mean and conclusion.

Response: Thanks to the reviewer for this valuable comment. Retrograde urethrography was performed to evaluate the diameter of the urethra in normal healthy rabbits and subgroup VI (PTB@SA-Rapamycin). As shown in Supplementary Fig. 37, the ratio of urethral diameter to individual femoral shaft diameter did not show a statistical difference between the above two groups, further suggesting that the anti-fibrotic catheter is effective in preventing stricture formation by modulating the tissue healing process.

Regarding the sample size, we sincerely accepted the reviewer's suggestion and supplemented animal experiments to increase the sample size of each group to 5. We have revised the relevant data (Fig. 6 and Supplementary Fig. 37~45) and the corresponding descriptions in the revised manuscript (Page 27~31). In addition, we collected optical photographs of urethral lesions in each group and integrated them in Fig. 6d. It can be observed that except for the PTB@SA-rapamycin group, the rabbit urethral injury developed into whitish, hard, wrinkled and even raised scars after 1 month in the other groups. However, in the anti-fibrotic catheter group, the urethral injury repaired well with smooth and rosy urethral mucosa. Overall, the results of animal experiments definitively demonstrate the efficacy and safety of the anti-fibrotic catheter in preventing stricture formation after urethral injury.

Figure S37. Comparison of urethral diameter in normal healthy rabbits and rabbits using PTB@SA-Rapamycin catheter after urethral injury. (a) Representative retrograde urethrogram of healthy rabbits. **(b)** Representative retrograde urethrogram of the rabbit in the anti-fibrotic catheter group. **(c)** The ratio of urethra diameter to femoral shaft diameter in normal healthy rabbits and rabbits in anti-fibrotic catheter (PTB@SA-Rapamycin) group. Data are presented as the mean \pm s.d.; n=5. ns means no statistical difference.

Comments to the author

The authors of this manuscript have done an excellent work designing a catheter covered with a protein nanofilm capable of preventing urethral stricture by prevention of bacterial colonization and controlled release of rapamycin. The authors examined the physical characteristics, antibacterial properties, and drug release functions of the protein nanofilm. By indwelling the hybrid coating-modified catheter in the urethra of adult male rabbits, the authors demonstrated that the catheter could effectively prevent the development of urethral stricture.

Generally, this work is innovative and great. A few comments are as follows:

Q1

In the first paragraph of the introduction, the author pointed out a clinical problem that current curative treatment options, such as urethrotomy, dilatation and urethroplasty, for urethral stricture could not achieve satisfactory results. This clinical problem exists in patients who have already developed severe urethral strictures. However, the animal model used in this study only simulated the acute urethral injury, and the results could only demonstrate the fact that PTB@SA-Rapamycin catheter was able to prevent the occurrence of urethral stricture. Therefore, the clinical problem raised in the introduction should be revised.

Response: Thanks to the reviewer for your helpful comments. We have revised the Introduction to clarify precisely the clinical concerns of this work. Urethral stricture is a common disease after urethral injury. The incidence of mucosal injury and secondary urethral strictures after various transurethral procedures, including traumatic catheterization, tends to increase with an aging population and advances in medical technology. Due to the complexity of its treatment, once the urethral injury develops into urethral stricture, it will pose challenges to both patients and urologists. This situation has also prompted researchers and clinicians to explore novel and clinically promising alternative or adjunctive strategies beyond surgery. In this work, targeting the well-defined pathological basis of ECM metabolism dysregulation in urethral strictures, our main intention is to combine a local anti-fibrotic drug delivery system with the urinary catheter to construct a novel drug-loaded device that can prevent the formation of urethral stricture by inhibiting fibrosis process. We apologize for the ambiguity caused by our inappropriate description and have revised the Introduction as follows to emphasize that this study addresses urethral strictures secondary to urethral injury (Page 2).

Urethral stricture is a common disease (200–1200 cases per 100000 individuals²²) following urethral injury. The pathological state of extracellular matrix (ECM) metabolic disorder caused by injury (including surgery, pelvic fracture, inflammatory injury and traumatic catheterization) results in the replacement of normal urethral tissue by dense fibers interspersed with fibroblasts²². This aggravated fibrosis eventually leads to progressive urethral lumen reduction consequent symptomatic dysuria and even renal impairment. With the growing demand for healthcare brought about by the aging population and advances in medical technology, the increasing incidence of mucosal injury and secondary urethral strictures caused by various transurethral procedures has attracted great attention^{23,24}. Especially in patients with spinal cord injury or in intensive care, urethral stricture or erosion due to long-term catheterization has been reported as high as 8.7%²⁵. Once urethral injury progresses to urethral stricture, subsequent treatment is extremely troublesome,

which will pose challenges to both patients and urologists. The long-term success rate of the most commonly used endoscopic treatment is only 20~30%²², and although urethroplasty has developed rapidly in recent years, fibrous scarring may still develop in the urethral submucosa after substitute surgery, and this open surgery is less suitable for the elderly and frail patients. Indeed, the pathological state of ECM metabolic disorder and subsequent scar repair is not be rectified regardless of the treatment²⁶. Therefore, there is an urgent need to develop alternative strategies other than surgery and attempt to focus on regulating healing by inhibiting fibrosis progression following urethral injury, thereby preventing urethral strictures.

Q2

Why did the authors choose rapamycin as the therapeutic component preventing the urethral stricture? A meta-analysis published in European Urology by Pang et al. showed that mitomycin C had the best effect on urethral stricture among drugs used in the included clinical trials. Compared with other drugs, what are the advantages and disadvantages of rapamycin in treating urethral stricture?

Response: Thanks to the reviewer for this comment. it should be pointed out that as a proof-of-concept study, the developed drug delivery system in this work is versatile and scalable. As shown in Supplementary Fig. 46, the PTB@SA nanofilm can also be used for long-term delivery of paclitaxel. Thus, it is feasible to encapsulate and immobilize other therapeutic agents using a similar strategy. In terms of preventing urethral strictures, we chose rapamycin as the therapeutic component of drug-loaded urinary catheters for the following two reasons:

- 1) Rapamycin (also known as sirolimus), which was initially used to prevent immune rejection response after transplantation, later also played an important role in inhibiting intimal hyperplasia and preventing coronary restenosis due to its favorable antiproliferative and antifibrotic effects. Inspired by this, our group verified the potential availability of rapamycin in inhibiting urethral stricture after urethral injury through a series of *in vitro* and *in vivo* experiments. For instance, after urethral injury, fibroblast proliferation and collagen deposition in rabbit urethral tissue were significantly inhibited by intraurethral perfusion of rapamycin for 4 weeks⁴¹. In addition, the cell proliferation and collagen synthesis of human urethral scar-derived fibroblasts were also inhibited in a dose-dependent manner under treatment with rapamycin⁴². These studies laid a solid research foundation for this work and encouraged us to further utilize rapamycin to construct a local drug delivery system for intervening urethral stricture formation.
- 2) In addition to efficacy, the safety of drug-loaded devices also has important implications for their clinical prospects. Admittedly, local application of mitomycin-C (MMC) showed encouraging results in the prevention and treatment of urethral stricture. However, as a cytotoxic chemotherapeutic agent, topical application of MMC to the mucosa has been reported to cause side effects such as fibrinous debris (which further causes airway obstruction), corneal edema and perforation, necrotizing scleritis and corneal ulceration^{43,44}. Moreover, a prospective study showed that 68% of subjects experienced mild to moderate urinary adverse events (including ureteral stenosis, hydronephrosis, vomiting, flank pain, hematuria and urinary tract infection) after using a sustained-release hydrogel polymer-based formulation containing MMC⁴⁵. Besides, according to a multi-institutional study published in the *Journal of Urology*, severe complications such as osteitis pubis, rectourethral fistula and necrosis of the bladder floor

occurred in 7% of patients undergoing endoscopic incision followed by MMC injection⁴⁶. In contrast, rapamycin, a commonly used and classic therapeutic component of coronary drug-eluting stents, has negligible side effects through topical application⁴⁷. The satisfactory outcome of rapamycin-eluting coronary stents in long-term clinical practice demonstrates the availability and safety of rapamycin in the field of local drug delivery, which also encouraged us to integrate rapamycin into the surface coating of the urinary catheter to achieve intraurethral drug therapy. The above discussion is condensed and supplemented in the revised manuscript (Page 30~31).

Q3

The author mentioned that the nanofilm surface exhibited abundant functional groups, which enabled the nanofilms to undergo ligand bonding, electrostatic interaction, hydrophobic interaction and hydrogen bonding with virtually arbitrary material, including metal, organic or inorganic materials. According to our experience, this property of the nanofilm makes it likely to interact with various organic and inorganic substances in urine and result in surfaces blockade, stone formation and catheter blockage. Is there any data showing the appearance of the hybrid nanofilm-coated catheters after indwelling for 30 days? We want to check if the surface of all catheters were still clear at the end of the experiment. In fact, bacterial contamination and stone formation are both thorny challenges in the development of functional urinary catheters.

Response: Thank the reviewer for this helpful comment. Encrustation is indeed a thorny problem for patients who require long-term indwelling catheters, which involves the deposition of inorganic salts on the surface of the device or the inner wall of the lumen, and further leads to pain, difficult removal of catheter, blockage of urinary flow and intractable bacterial biofilms. Two important aspects that promote the development of crusts include supersaturation of crystals in the urine and bacterial infection. However, due to the unique calcium metabolism of rabbits, that is, almost all calcium in the diet can be absorbed while the excess calcium is excreted through the kidney to maintain blood calcium stability, the calcium in rabbit urine is affected by the combination of diet, water intake, physical activity and other hormones^{48,49}. To better exclude the influence of irrelevant variables on the results, we performed a well-established *in vitro* encrustation experiment to compare the formation of urinary encrustations on bare and PTB@SA-modified urinary catheters in artificial urine, artificial urine in the presence of *P. mirabilis* or human urine⁵⁰. As shown in Supplementary Fig. 20 a and b, there was no difference in encrustation weight between the control and PTB@SA-coated samples exposed to the same urine environment at each time point, except that the weight of crystals formed in human urine on day 3 was smaller in the PTB@SA group. At the same time point, the presence of *P. mirabilis* significantly aggravated the encrustation, because the urease produced by *P. mirabilis* would increase the pH of urine and promote the deposition of calcium phosphate and magnesium⁵¹. After 30 days, both the bare catheter and the PTB@SA coated surfaces were covered with cauliflower-like smooth deposits and interspersed with large coffin-like crystals (Supplementary Fig.20 c). Furthermore, we performed Energy Dispersive Spectroscopy (EDS) analysis of these deposits and determined that the smooth layer was mainly composed of Ca, C, P and O, presumably being calcium carbonate apatite crystallites. while the large coffin-shaped crystals were composed of Mg, O and P, which could be assigned to struvite (magnesium phosphate) (Supplementary Fig.20 d-i). Overall, the PTB@SA nanofilm did not induce more severe encrustation formation on the device surface within 1 month, indicating that it would not pose an

additional threat when applied to construct functional urinary catheters. Corresponding results and discussions have been incorporated into the revised manuscript (Page 16) and supporting information (Page S15~S16).

It is worth pointing out that since the PTB@SA coating has a balanced surface charge distribution similar to that of zwitterionic polymers, it can effectively avoid bacterial adhesion and biofilm formation, which may contribute to reduce encrustation caused by *P. mirabilis* adhesion and its subsequent migration along the catheter surface⁴. However, in this in vitro encrustation model, the coating failed to kill *P. mirabilis* present in the device and thus exhibited a similar encrustation propensity to the unmodified catheter. As stated by the reviewers, anti-encrustation is an important challenge for medical devices such as urinary catheters and ureteral stents. Although the performance of the coating reported in this work in reducing crusts is not superior enough, this will become the focus of our next research work.

Figure S20. *In vitro* evaluation of the encrustation performance of bare silicone catheters and the PTB@SA nanofilm-modified catheters. (a) The weight of crystals deposited on bare catheters or the PTB@SA coated catheters after exposure to artificial urine, artificial urine containing *Proteus mirabilis* (*P. mirabilis*), or human urine. Data are presented as the mean \pm s.d; n=4. * P<0.05. **(b)** The weight of crystals deposited on bare catheters or the PTB@SA coated catheters during the first week. Data are presented as the mean \pm s.d; n=4. * P<0.05. **(c)** SEM of bare catheter and the

PTB@SA coating after 30 days of encrustation experiment *in vitro*. (d) SEM image of bare catheter surface after encrustation experiment. (e, f) EDS analysis of area A (e) and B (f) in Fig S20 d. (g) SEM image of the PTB@SA coating surface after encrustation experiment. (h, i) EDS analysis of area A (h) and B (i) in Fig S20 g.

Q4

The nanofilm showed a consistent WCA approximately 80°. This means the nanofilm-coated catheter is not very hydrophilic and has higher friction with urethral. Was lubricant used during indwelling catheters. If lubricant was used, were there any experiments investigating if the lubricant had any effect on the stability of the nanofilm and the controlled drug release? Catheters for clinical use are often covered with super hydrophilic film or lubricants to avoid urethral injury and inflammation. Consistently, the authors observed that the pristine urinary catheter caused edema and necrosis in rabbits' urethral mucosa after indwelling for 1 month. Parameters about the friction should be compared among pristine urinary catheter and nanofilm coated catheters.

Response: Thanks to the reviewer for this helpful comment. The abundant functional groups exposed on the nanofilm surface, especially the sodium alginate contained therein, allow the coating to interact with water molecules to hydrate and lubricate the catheter to a certain extent. Furthermore, the coefficient of friction (COF) of the saline-wetted unmodified urinary catheter and the PTB@SA-coated urinary catheter was characterized by a coefficient of friction meter. As shown in Figure R2, the COF of the PTB@SA membrane was lower than that of the unmodified catheter. Therefore, in all animal experiments involving indwelling urinary catheters, the catheters were lubricated with saline prior to catheterization. This organic-free physiological solution did not affect the controlled release of the encapsulated drug, and theoretically other water-soluble lubricants are also suitable. The urethral implantation experiment (Supplementary Fig. 19) and H&E staining of the uninjured urethra in each group in the urethral injury experiment (Supplementary Fig. 38) showed that catheterization performed by this method and indwelling catheter for one month did not cause edema, injury or necrosis in the urethral mucosa and submucosa, indicating that the functionalized urinary catheter has good histocompatibility and practical prospects, while the urethral edema and inflammation shown in Fig. 3 in the control catheter group were mainly due to bacterial urinary tract infection caused by the dense bacterial biofilm on the pristine urinary catheter surface.

Figure R2. Friction coefficients of bare urinary catheter and the PTB@SA nanofilm-modified catheter. Data are presented as the mean \pm s.d; n=3. *** P<0.001.

Q5

Reference should be added for the sentence “... the PTB@SA nanofilm still exhibited a low negative surface zeta potential, and thus may keep a good anti-biofilm ability.”

Response: Thank the reviewer for this comment. Regarding the reason for the excellent anti-biofilm performance of the PTB@SA nanofilm, we supplied the relevant discussion in page 15 and cite reference as Ref. 43, which demonstrates the formation of ultra-low fouling peptides from certain natural amino acids based on the mixed charge design principle. Similarly, the balanced distribution of charged amino acids on the nanofilm surface in this work can also form a hydration layer on the surface of the material, which acts as a barrier against bacterial adhesion and thus resists biofilm formation.

Q6

We noticed that four rabbits were in each group for the evaluation of the antibacterial biofilm performance of the nanofilm and three rabbits were in each group evaluate the effect of nanofilm on urethral stricture. How did the authors decide the sample size for each experiment? Why did experiments with more complex surgical procedures (more likely to be biased) have smaller sample sizes?

Response: Thank the reviewer for this comment. For the complex animal model such as urethral injury, in order to better confirm the efficacy of the anti-fibrotic catheter, we supplemented the animal experiment to increase the sample size of each group to 5. Representative images of retrograde urethrography are shown below (Fig.R1). In addition, we collected optical photographs of urethral lesions in each group and integrated them in Fig. 6d. It can be observed that except for the PTB@SA-rapamycin group, the rabbit urethral injury developed into whitish, stiff, wrinkled and even raised scars after 1 month in the other groups. However, in the anti-fibrotic catheter group, the urethral injury repaired well with smooth and ruddy urethral mucosa. We have revised the relevant data (Fig. 6 and Supplementary Fig. 37~45) and the corresponding descriptions in the revised manuscript (Page 27~31). Overall, the results of animal experiments definitively demonstrate the efficacy and safety of the anti-fibrotic catheter in preventing stricture formation after urethral injury.

Figure R1. Representative images of retrograde urethrography after 30 days. The yellow arrow indicates the coagulation site.

Figure. 6 The sustained-release coating of rapamycin on the urinary catheter attenuates collagen deposition and inhibits the formation of strictures post urethral injury.

a, Representative retrograde urethrogram of rabbits with different treatments after urethral electrocoagulation, using 76% Meglumine Diatrizoate as a contrast agent. **b**, Quantitative analysis of urethral lumen reduction and submucosal collagen density in rabbits with different treatments.

Data are mean \pm s.d.; n=5. **c**, Corresponding results of the PTB@SA-Rapamycin catheter compared with other preclinical studies of topical therapy to suppress urethral stricture. **d**, Gross specimen, H&E staining and immunohistochemical staining with anti-alpha smooth muscle actin antibody (α -SMA, brown), anti-matrix metalloproteinase 1 antibody (MMP1, brown), and type I collagen antibody (Collagen I, green) of the urethral injury sites at 1 month. Nuclei (blue) were stained with DAPI. Yellow triangles indicate urethral scar tissue. The asterisks represent urethral luminal side, and the dotted lines indicate the boundary between the submucosa and the muscularis. **e**, Quantification of α -SMA, MMP1 positive area and Collagen I fluorescence intensity in **Fig. 6d**. Data are mean \pm s.d.; n=5. **f, g**, Hematological examination to show the changes of alanine aminotransferase (ALT, **f**) and aspartate aminotransferase (AST, **g**) with different treatments for 1 month. Data are mean \pm s.d.; n=5. **h**, H&E staining of rabbit liver after systemic administration or indwelling rapamycin sustained-release catheter for 1 month. (**a, d**) **i**, Control, **ii**; Unmodified catheter; **iii**, Systemic administration; **iv**, PTB@SA; **v**, Burst-releasing; **vi**, PTB@SA-Rapamycin. * P<0.05, ** P<0.01, *** P<0.001, **** P<0.0001, ns means no statistical difference.

Q7

The authors cannot directly compare the results of independent experiments performed by different operators under different experimental conditions (such as surgical instruments, degree of urethral injury and therapy duration) to conclude that the effect of nanofilm in this study was better than that of drugs reported in other studies.

Response: Thank the reviewer for this comment. We fully agree with the reviewer's comments and supplied the following description to the Methods (Page 43). In order to exclude the interference of various experimental conditions to more objectively compare the preventive effect on urethral stricture reported in independent studies, we reprocessed the original data from each study and calculated the mean percent improvement in lumen stenosis for each intervention according to the following formula to quantify the improvement in severity of urethral strictures with different treatments (Fig. 6c), Where Lumen reduction(Control) represents the reduction degree of the urethral lumen in the control group, and Lumen reduction(Experiment) represents the reduction degree of the urethral lumen in the experimental group.

$$\text{Mean percent improvement} = \frac{\text{Lumen reduction}(\text{Control}) - \text{Lumen reduction}(\text{Experiment})}{\text{Lumen reduction}(\text{Control})} \times 100\%$$

The relevant information of each study, including the drug used, the method of administration, and the outcomes of the experimental and control group, are all listed in detail in Table S5.

Table S5. Overview of preclinical outcomes of topical therapy to prevent urethral strictures.

Study	Topical intervention	Lumen reduction		Mean percent improvements in lumen stenosis
		Control group	Experimental group	
Kurt 2017 ^[7]	40 mg Triamcinolone, submucosal injection, once	91.0%	49.0%	46.2%
Chong 2011 ^[8]	0.1 mg Rapamycin, urethral irrigation, daily for 28 days	69.1%	56.6%	18.1%
	1 mg Rapamycin, urethral irrigation, daily for 28 days	69.1%	36.5%	47.2%
Kurt 2017 ^[7]	0.5 mg/mL Mitomycin-C (MMC), hydropathic compress, once	91.0%	45.0%	50.5%
Shinchi 2019 ^[9]	0.2 mg Insulin-like growth factor 1 (IGF-1), impregnated collagen sutured to the catheter, catheterization for 14 days	79.8%	39.3%	50.8%
Fu 2014 ^[10]	0.01 mg Docetaxel, urethral irrigation, daily for 28 days	84.7%	48.0%	43.3%
	0.1 mg Docetaxel, urethral irrigation, daily for 28 days	84.7%	36.9%	56.4%
This research	3 mg Rapamycin, sustained release coating modified catheter, catheterization for 30 days	51.8%	10.6%	79.2%

In summary, by comparing the mean percent improvement in lumen stenosis between this study and the results reported in other literatures, rather than the value of urethral lumen in independent studies, the influence of experimental conditions on the comparison results can be offset to a certain extent. The mean percent improvement in luminal stenosis with the anti-fibrotic catheter was as high as 79.2%, suggesting a potential advantage of the PTB@SA-rapamycin catheter in the prevention of urethral strictures compared with other topical dosage regimens.

Q8

In the figure legend of figure 6, the sentence “(b, d, e) i, Control, ii; Unmodified catheter; iii,...” should be revised. Only a and d are numbered. Please check the use of commas and semicolons.

Response: Thank the reviewer for this comment, and we are sorry for this error in the legend. We have made revisions in the revised manuscript.

Q9

At least, there are some errors in references 56. Some authors are missing and the original title should be “Sirolimus-eluting Biodegradable Poly-L-Lactic Acid Stent to Suppress Granulation Tissue Formation in the Rat Urethra.”

Response: Thank the reviewer for this comment. We have corrected this reference as follows and proofread all references cited in the article.

“Kim, K. Y. *et al.* Sirolimus-eluting Biodegradable Poly-L-Lactic Acid Stent to Suppress Granulation Tissue Formation in the Rat Urethra. *Radiology* **286**, 140–148 (2017).”

References :

1. Nilles, J., Weiss, J. & Theile, D. Crystal violet staining is a reliable alternative to bicinchoninic acid assay-based normalization. *Biotechniques* **73**, 131–135 (2022).
2. Riool, M. *et al.* Staphylococcus epidermidis originating from titanium implants infects surrounding tissue and immune cells. *Acta Biomater.* **10**, 5202–5212 (2014).
3. Gu, J., Su, Y., Liu, P., Li, P. & Yang, P. An Environmentally Benign Antimicrobial Coating Based on a Protein Supramolecular Assembly. *ACS Appl. Mater. Interfaces* **9**, 198–210 (2017).
4. Ramstedt, M. *et al.* Evaluating Efficacy of Antimicrobial and Antifouling Materials for Urinary Tract Medical Devices: Challenges and Recommendations. *Macromol. Biosci.* **19**, 1–26 (2019).
5. Solano, C., Echeverz, M. & Lasa, I. Biofilm dispersion and quorum sensing. *Curr. Opin. Microbiol.* **18**, 96–104 (2014).
6. Zhang, N. *et al.* Different types of T-effector cells orchestrate mucosal inflammation in chronic sinus disease. *J. Allergy Clin. Immunol.* **122**, 961–968 (2008).
7. Uthamanthil, R. K. *et al.* Urinary catheterization of male rabbits: A new technique and a review of urogenital anatomy. *Journal of the American Association for Laboratory Animal Science* vol. 52 180–185 at (2013).
8. Hua, X. *et al.* An Experimental Model of Anterior Urethral Stricture in Rabbits With Local Injections of Bleomycin. *Urology* **116**, 230.e9-230.e15 (2018).
9. Abizaïd, A. Sirolimus-eluting coronary stents: A review. *Vasc. Health Risk Manag.* **3**, 191–201 (2007).
10. Rashidbenam, Z. *et al.* Overview of Urethral Reconstruction by Tissue Engineering: Current Strategies, Clinical Status and Future Direction. *Tissue Eng. Regen. Med.* **16**, 365–384 (2019).
11. Ueshima, E. *et al.* Macrophage-secreted TGF- β 1 contributes to fibroblast activation and ureteral stricture after ablation injury. *Am. J. Physiol. - Ren. Physiol.* **317**, F52–F64 (2019).
12. Chapman, D., Kinnaird, A. & Rourke, K. Independent Predictors of Stricture Recurrence Following Urethroplasty for Isolated Bulbar Urethral Strictures. *J. Urol.* **198**, 1107–1112 (2017).
13. Kurt, O. *et al.* Effect of Tadalafil on Prevention of Urethral Stricture after Urethral Injury: An Experimental Study. *Urology* **91**, 243.e1-243.e6 (2016).
14. Yoshizaki, A. *et al.* Treatment with rapamycin prevents fibrosis in tight-skin and bleomycin-induced mouse models of systemic sclerosis. *Arthritis Rheum.* **62**, 2476–2487 (2010).
15. Rostaing, L. & Kamar, N. mTOR inhibitor/proliferation signal inhibitors: entering or leaving the field? *J. Nephrol.* **23**, 133–142 (2010).
16. N., F.-R. One-Year Results for the ROBUST III Randomized Controlled Trial Evaluating the Optilume® Drug-Coated Balloon for Anterior Urethral Strictures. Letter. *J. Urol.* **207**, 866–875 (2022).
17. Guzmán-Esquivel, J. *et al.* Metalloproteinase-1 usefulness in urethral stricture treatment. *Int. Urol. Nephrol.* **43**, 763–769 (2011).
18. Shinchi, M. *et al.* Insulin-like growth factor 1 sustained-release collagen on urethral catheter prevents stricture after urethral injury in a rabbit model. *Int. J. Urol.* **26**, 572–577 (2019).
19. Kurt, O. *et al.* Effect of mitomycin - C and triamcinolone on preventing urethral strictures. *Int. Braz J Urol* **43**, 939–945 (2017).
20. Fu, D., Chong, T., Li, H., Zhang, H. & Wang, Z. Docetaxel inhibits urethral stricture formation,

- an initial study in rabbit model. *PLoS One* **9**, 1–6 (2014).
21. Chong, T. *et al.* Rapamycin inhibits formation of urethral stricture in rabbits. *J. Pharmacol. Exp. Ther.* **338**, 47–52 (2011).
 22. Hampson, L. A., McAninch, J. W. & Breyer, B. N. Male urethral strictures and their management. *Nat. Rev. Urol.* **11**, 43–50 (2014).
 23. Lumen, N. *et al.* Etiology of Urethral Stricture Disease in the 21st Century. *J. Urol.* **182**, 983–987 (2009).
 24. Chen, M. L., Correa, A. F. & Santucci, R. A. Urethral Strictures and Stenoses Caused by Prostate Therapy. *Rev. Urol.* **18**, 90–102 (2016).
 25. Hollingsworth, J. M. *et al.* Determining the Noninfectious Complications of Indwelling Urethral Catheters. *Ann. Intern. Med.* **159**, 401–410 (2013).
 26. Prihadi, J. C., Sugandi, S., Siregar, N. C., Soejono, G. & Harahap, A. Imbalance in extracellular matrix degradation in urethral stricture. *Res. Reports Urol.* **10**, 227–232 (2018).
 27. Sievert, K. D. *et al.* Introducing a large animal model to create urethral stricture similar to human stricture disease: A comparative experimental microscopic study. *J. Urol.* **187**, 1101–1109 (2012).
 28. Delai, F. *et al.* Development and characterization of urethral stricture model in rabbits. *J. Med. Coll. PLA* **25**, 351–358 (2010).
 29. Hirano, Y. *et al.* Myofibroblast-dominant proliferation associated with severe fibrosis in bulbar urethral strictures. *Int. J. Urol.* (2022) doi:<https://doi.org/10.1111/iju.15053>.
 30. Sanyal, A. J. *et al.* The Natural History of Advanced Fibrosis Due to Nonalcoholic Steatohepatitis: Data From the Simtuzumab Trials. *Hepatology* **70**, 1913–1927 (2019).
 31. Hu, X. *et al.* Amyloid-Like Protein Aggregates: A New Class of Bioinspired Materials Merging an Interfacial Anchor with Antifouling. *Adv. Mater.* **32**, 1–11 (2020).
 32. Zhang, F. *et al.* Stretchable and biocompatible bovine serum albumin fibrous films supported silver for accelerated bacteria-infected wound healing. *Chem. Eng. J.* **417**, 129145 (2021).
 33. Ouyang, J. *et al.* A facile and general method for synthesis of antibiotic-free protein-based hydrogel: Wound dressing for the eradication of drug-resistant bacteria and biofilms. *Bioact. Mater.* **18**, 446–458 (2022).
 34. Chang, M. *et al.* Suppression of Sunscreen Leakage in Water by Amyloid-like Protein Aggregates. *ACS Appl. Mater. Interfaces* **13**, 42451–42460 (2021).
 35. Liu, Y., Tao, F., Miao, S. & Yang, P. Controlling the Structure and Function of Protein Thin Films through Amyloid-like Aggregation. *Acc. Chem. Res.* **54**, 3016–3027 (2021).
 36. Li, C., Xu, L., Zuo, Y. Y. & Yang, P. Tuning protein assembly pathways through superfast amyloid-like aggregation. *Biomater. Sci.* **6**, 836–841 (2018).
 37. Louis-Jeune, C., Andrade-Navarro, M. A. & Perez-Iratxeta, C. Prediction of protein secondary structure from circular dichroism using theoretically derived spectra. *Proteins Struct. Funct. Bioinforma.* **80**, 374–381 (2012).
 38. Venyaminov, S. Y., Baikalov, I. A., Wu, C. S. C. & Yang, J. T. Some problems of CD analyses of protein conformation. *Anal. Biochem.* **198**, 250–255 (1991).
 39. Chen, S., Cao, Z. & Jiang, S. Ultra-low fouling peptide surfaces derived from natural amino acids. *Biomaterials* **30**, 5892–5896 (2009).
 40. Wang, F. *et al.* Review of the research on anti-protein fouling coatings materials. *Prog. Org. Coatings* **147**, 105860 (2020).

41. Chong, T. *et al.* Rapamycin inhibits formation of urethral stricture in rabbits. *J. Pharmacol. Exp. Ther.* **338**, 47–52 (2011).
42. Fu, D. *et al.* Rapamycin Inhibits the Growth and Collagen Production of Fibroblasts Derived from Human Urethral Scar Tissue. *Biomed Res. Int.* **2018**, 1–9 (2018).
43. McCurdy Hueman, E. & Simpson, C. B. Airway complications from topical mitomycin C. *Otolaryngol. - Head Neck Surg.* **133**, 831–835 (2005).
44. Cinik, R. *et al.* The Effect of Everolimus on Scar Formation in Glaucoma Filtering Surgery in a Rabbit Model. *Curr. Eye Res.* **41**, 1438–1446 (2016).
45. Porta, C. *et al.* An evaluation of UGN-101, a sustained-release hydrogel polymer-based formulation containing mitomycin-C, for the treatment of upper urothelial carcinomas. *Expert Opin. Pharmacother.* **21**, 2199–2204 (2020).
46. Redshaw, J. D. *et al.* Intralesional injection of mitomycin C at transurethral incision of bladder neck contracture may offer limited benefit: TURNS study group. *J. Urol.* **193**, 587–592 (2015).
47. Roos, J. C. P. & Murthy, R. Sirolimus (rapamycin) for the targeted treatment of the fibrotic sequelae of Graves' orbitopathy. *Eye* **33**, 679–682 (2019).
48. Redrobe, S. Calcium metabolism in rabbits. *Semin. Avian Exot. Pet Med.* **11**, 94–101 (2002).
49. Clauss, M. *et al.* Influence of diet on calcium metabolism, tissue calcification and urinary sludge in rabbits (*Oryctolagus cuniculus*). *J. Anim. Physiol. Anim. Nutr. (Berl.)* **96**, 798–807 (2012).
50. Jones, D. S., Djokic, J. & Gorman, S. P. Characterization and optimization of experimental variables within a reproducible bladder encrustation model and in vitro evaluation of the efficacy of urease inhibitors for the prevention of medical device-related encrustation. *J. Biomed. Mater. Res. - Part B Appl. Biomater.* **76**, 1–7 (2006).
51. Yao, Q. *et al.* Bio-inspired antibacterial coatings on urinary stents for encrustation prevention. *J. Mater. Chem. B* **10**, 2584–2596 (2021).

Reviewers' Comments:

Reviewer #1:

Remarks to the Author:

I recognized that the authors made proper efforts for revision. The revised manuscript becomes acceptable.

Reviewer #3:

Remarks to the Author:

The authors revised manuscript thoroughly and carefully. The manuscript is ready for publication.

Reviewer #4:

Remarks to the Author:

Thanks for your kind reply and revision.

Reviewer #5:

Remarks to the Author:

This is a tremendous body of work, very meticulous and thoughtful. Most of the responses were in-depth and most of the requested changes were made. I would like to congratulate the authors with completion of this project. There are only 3 additional comments:

1) in response to Reviewer#2 there is no direct explanation to the question #4 on what the reason is to sacrifice the animals so early after catheter removal. In real life it takes weeks for strictures to mature and appear clinically significant. Other animal models of stricture disease use a minimum of 4 weeks to 6 months before declaring a resolution/non-formation of stricture. In humans this time is even longer (no one believes success if it is reported less than 8-12 months since the treatment. I understand that authors saw all kind of indirect evidence and hopeful signs of no stricture formation in treatment group, but at 2 weeks these findings have no clinical significance.

2) Why is catheter staying for 4 weeks? under what clinical circumstances a urethral catheter would stay for that long? Not after transurethral procedures (3-7 days typically), not after traumatic catheter (7-14 days), not after prostatectomy (3-14 days). Why 4 weeks?

3) in many instances the retrograde urethrograms are shown in wrong orientation. Typically, distal portion of urethra is either shown facing 3 or 9 o'clock position (consistently through all images), cephalad is at 12 o'clock, caudad is at 6 o'clock. At least in S37 and in figure 6 the orientation is unknown and this makes these figures very confusing. Additionally, some of the radiographs are darker and with poor contrast making it difficult to assess urethral lumen, I would recommend revising these images to have consistent orientation and brightness/contrast.

Dear Reviewers,

We sincerely appreciate your valuable comments. We have made the revisions to all the comments in the revised manuscript (marked in red color) and supporting information (SI). The following is the point-to-point response to the reviewers' comments.

This is a tremendous body of work, very meticulous and thoughtful. Most of the responses were in-depth and most of the requested changes were made. I would like to congratulate the authors with completion of this project. There are only 3 additional comments:

1) in response to Reviewer#2 there is no direct explanation to the question #4 on what the reason is to sacrifice the animals so early after catheter removal. In real life it takes weeks for strictures to mature and appear clinically significant. Other animal models of stricture disease use a minimum of 4 weeks to 6 months before declaring a resolution/non-formation of stricture. In humans this time is even longer (no one believes success if it is reported less than 8-12 months since the treatment. I understand that authors saw all kind of indirect evidence and hopeful signs of no stricture formation in treatment group, but at 2 weeks these findings have no clinical significance.

Response: Thanks to the reviewer for this valuable comment, we will further explore the long-term effects of the anti-fibrotic catheter in our future experiments. In response to this comment, we supplemented this limitation in the corresponding section of Results and Discussion on page 26 as follows: "The above results suggest that the anti-fibrotic catheter has shown impressive preliminary results in retarding the progression of urethral injuries to strictures, but do have a limitation that its long-term effect (six months to one year) in inhibiting urethral strictures need to be clarified in future experiments." In addition, in the Summary section (page 31), we have revised the wording as follows to make the statement more rigorous and objective. "In this regard, the promotion of PTB@SA-Rapamycin catheters as an alternative to conventional catheters is expected to be of great benefit to numerous people at high risk for urethral strictures (eg, undergoing transurethral intervention, traumatic catheterization or urethroplasty)."

2) Why is catheter staying for 4 weeks? under what clinical circumstances a urethral catheter would stay for that long? Not after transurethral procedures (3-7 days typically), not after traumatic catheter (7-14 days), not after prostatectomy (3-14 days). Why 4 weeks?

Response: Thanks to the reviewer for this valuable comment. As mentioned by the reviewer, the intervention time used in the animal experiment should take into account the actual clinical needs. The specific reasons for indwelling the catheter for 1 month after urethral injury are stated as follows.

1) Due to the unique anatomical structure of the urethra and the stimulating effect of urine, the healing time of urethral wounds is longer than that of skin tissue, with each healing stage lagging behind that of skin. It has been reported that rat urethral injury ends the proliferative phase characterized by fibroblast proliferation activation and angiogenesis on the 10th day¹, and the subsequent maturation and remodeling phase involving the arrangement of collagen and other connective tissues around the urethra is a key stage of urethral fibrosis². Therefore, in order to effectively inhibit urethral stricture, the drug release period should completely cover all stages of urethral injury healing. Our previous studies have shown that continuous transurethral infusion of anti-fibrotic drugs (such as rapamycin or docetaxel) for 28 days after urethral injury can effectively

inhibit urethral stricture^{3,4}, and other experimental studies on the prevention of urethral strictures by topical administration also commonly used 1 month as the duration of drug action⁵⁻⁷.

2) According to the *Campbell-Walsh urology* and European Association of Urology (EAU) guidelines on urologic trauma^{8,9}, partial posterior urethral rupture require catheterization to drain urine and provide a support surface for mucosal repair until the injury heals. After urethral reunion, catheterization for 4-6 weeks is also recommended. For urethral injuries that cannot be treated conservatively, EAU recommends catheterization for 2-3 weeks after urethroplasty. Based on the above, in order to assess the promise of the anti-fibrotic catheter in inhibiting the progression of urethral injury to urethral strictures and preventing recurrent strictures at the incision site of urethroplasty, we evaluated its efficacy and safety after 1 month of indwelling.

In summary, in the consideration of the pathophysiology of urethral injury healing and the current clinical management standards, and also combining with literature research and our previous studies, we set 1 month as the intervention time and proved the beneficial effect of the anti-fibrotic catheter in inhibiting urethral strictures.

3) in many instances the retrograde urethrograms are shown in wrong orientation. Typically, distal portion of urethra is either shown facing 3 or 9 o'clock position (consistently through all images), cephalad is at 12 o'clock, caudad is at 6 o'clock. At least in S37 and in figure 6 the orientation is unknown and this makes these figures very confusing. Additionally, some of the radiographs are darker and with poor contrast making it difficult to assess urethral lumen, I would recommend revising these images to have consistent orientation and brightness/contrast.

Response: Thanks to the reviewer for this helpful suggestion. In order to make the urethrography images clearer, we unified the display directions of all urethrography images in this paper so that the cephalad is at 12 o'clock and the caudal side is at 6 o'clock. In addition, the brightness and contrast of some images have been adjusted. We have updated all relevant figures in the manuscript and supporting information (Figure 6, FigureS37 and Figure S40).

Figure. 6 The sustained-release coating of rapamycin on the urinary catheter attenuates collagen deposition and inhibits the formation of strictures post urethral injury.

a, Representative retrograde urethrogram of rabbits with different treatments after urethral electrocoagulation, using 76% Meglumine Diatrizoate as a contrast agent. **b**, Quantitative analysis of urethral lumen reduction and submucosal collagen density in rabbits with different treatments. Data are mean \pm s.d.; $n=5$. **c**, Corresponding results of the PTB@SA-Rapamycin catheter compared

with other preclinical studies of topical therapy to suppress urethral stricture. **d**, Gross specimen, H&E staining and immunohistochemical staining with anti-alpha smooth muscle actin antibody (α -SMA, brown), anti-matrix metalloproteinase 1 antibody (MMP1, brown), and type I collagen antibody (Collagen I, green) of the urethral injury sites at 1 month. Nuclei (blue) were stained with DAPI. Yellow triangles indicate urethral scar tissue. The asterisks represent urethral luminal side, and the dotted lines indicate the boundary between the submucosa and the muscularis. **e**, Quantification of α -SMA, MMP1 positive area and Collagen I fluorescence intensity in Fig. 6d. Data are mean \pm s.d.; n=5. **f**, **g**, Hematological examination to show the changes of alanine aminotransferase (ALT, **f**) and aspartate aminotransferase (AST, **g**) with different treatments for 1 month. Data are mean \pm s.d.; n=5. **h**, H&E staining of rabbit liver after systemic administration or indwelling rapamycin sustained-release catheter for 1 month. (a, d) i, Control; ii, Unmodified catheter; iii, Systemic administration; iv, PTB@SA; v, Burst-releasing; vi, PTB@SA-Rapamycin. * $P < 0.05$, ** $P < 0.01$, *** $P < 0.001$, **** $P < 0.0001$, ns means no statistical difference.

Figure S37. Comparison of urethral diameter in normal healthy rabbits and rabbits using PTB@SA-Rapamycin catheter after urethral injury. (a) Representative retrograde urethrogram of healthy rabbits. (b) Representative retrograde urethrogram of the rabbit in the anti-fibrotic catheter group. (c) The ratio of urethra diameter to femoral shaft diameter in normal healthy rabbits and rabbits in anti-fibrotic catheter (PTB@SA-Rapamycin) group. Data are presented as the mean \pm s.d.; n=5. ns means no statistical difference.

Figure S40. Rabbit urethral stricture in unmodified urinary catheter group and PTB@SA-Rapamycin group on the 14th day after removal of catheter. (a) Representative retrograde urethrograms using 76% meglumine diatrizoate as contrast agent. (b) Lumen reduction immediately after catheter removal or 2 weeks later as determined by retrograde urethrography. Data are mean \pm s.d.; n=5. (c) Comparison of gross specimens, H&E staining, Masson staining and α -SMA expression between unmodified urinary catheter group and PTB@SA-Rapamycin group at 14 days after catheter removal. Yellow triangles indicate urethral scar tissue, and the dotted lines indicate the boundary between the submucosa and the muscularis. (d, e) Quantitative results of collagen density and α -SMA positive area in **Figure S40c**. Data are mean \pm s.d.; n=5. *** P<0.001.

References:

1. Ninan, N., Thomas, S. & Grohens, Y. Wound healing in urology. *Adv. Drug Deliv. Rev.* **82–83**, 93–105 (2015).
2. Prihadi, J. C., Sugandi, S., Siregar, N. C., Soejono, G. & Harahap, A. Imbalance in extracellular matrix degradation in urethral stricture. *Res. Reports Urol.* **10**, 227–232 (2018).
3. Chong, T. *et al.* Rapamycin inhibits formation of urethral stricture in rabbits. *J. Pharmacol. Exp. Ther.* **338**, 47–52 (2011).
4. Fu, D., Chong, T., Li, H., Zhang, H. & Wang, Z. Docetaxel inhibits urethral stricture formation, an initial study in rabbit model. *PLoS One* **9**, 1–6 (2014).
5. Wang, Z., Li, Q., Wang, P. & Yang, M. Biodegradable drug-eluting urethral stent in limiting urethral stricture formation after urethral injury: An experimental study in rabbit. *J. Bioact. Compat. Polym.* **35**, 378–388 (2020).
6. Shin, J. H. *et al.* Tissue hyperplasia: Influence of a paclitaxel-eluting covered stent - Preliminary study in a canine urethral model. *Radiology* **234**, 438–444 (2005).
7. Wang, L. *et al.* Electrospun nanoyarn and exosomes of adipose-derived stem cells for urethral regeneration: Evaluations in vitro and in vivo. *Colloids Surfaces B Biointerfaces* **209**, 112218 (2022).
8. Partin, A. W., Wein, A. J., Kavoussi, L. R., Peters, C. A. & Dmochowski, R. R. *Campbell-Walsh urology*. (Elsevier Health Sciences, 2020).
9. Lumen, N. *et al.* Review of the current management of lower urinary tract injuries by the EAU trauma guidelines panel. *Eur. Urol.* **67**, 925–929 (2015).

Reviewers' Comments:

Reviewer #5:

Remarks to the Author:

All questions answered, changes made as requested.